# Multiscale Co-Manifold Learning on Tensors

## Abstract

Nonlinear manifold learning on tensors typically performs mode-wise embeddings that fail to capture implicit couplings between the geometries of the different modes. We propose a new framework called tensor co-manifold learning (TCML). TCML is designed to recover coupled low-dimensional structures *simultaneously* across all modes of multiway data (i.e., tensors or multi-dimensional arrays) and generalizes recent methods for co-manifold learning to higher-order tensors via a tensor-based multiscale approach to co-organizing rows, columns, and higher modes. By imposing smoothness constraints at various levels of granularity, we formulate a family of optimization problems that characterize smoothness across coarse-to-fine scales. We demonstrate that these problems are efficiently solvable and their solutions yield a multiscale distance between tensor slices along a given mode. These distances take into account the structure of the data along the other modes. We demonstrate how to utilize this multiscale distance measure to compute nonlinear embeddings of the data. The resulting embeddings are demonstrably more effective at revealing low-dimensional coupled structure than linear factorizations or nonlinear embeddings obtained by treating each mode independently.

## 1 Introduction

Multiway arrays, or tensors, are widespread in scientific and engineering applications involving complex heterogeneous data. For example, data in a neurogenomics study of brain development consists of a 3-way array of expression level measurements indexed by gene, space, and time (Liu et al., 2017). Other examples of 3-way data arrays consisting of matrices collected over time include video, email communications, internet network traffic, neuroscience, and more general multi-dimensional arrays, e.g., in -omics (Papalexakis et al., 2013; Sun et al., 2006; Mishne et al., 2016; Williams et al., 2018; Shahid et al., 2019; Schreiber et al., 2020). Tensor analysis also plays a role in fundamental machine learning and signal processing tasks, including compression, completion, parameter approximation, visualization, and recovering latent embeddings Shahid et al. (2019); Nie et al. (2020); Cai et al. (2023). Although tensors are a generalization of matrices, directly applying matrix methods after mode-wise matricization can discard multiway structure. Moreover, tensor rank, tensor decompositions, identifiability, and computation differ substantially from their matrix analogues (Acar and Yener, 2009; Kolda and Bader, 2009; Anandkumar et al., 2014; Cichocki et al., 2015; Sidiropoulos et al., 2017). These differences motivate methods that operate on the multiway structure directly (Mishne et al., 2016; Shahid et al., 2019; Stanley et al., 2020; Nie et al., 2020). In this paper, we are interested in understanding how to perform nonlinear dimension reduction for tensor data along each mode while taking into account the geometry of the other modes.

Nonlinear dimension reduction, or manifold learning, is a key step in many data analysis applications, such as exploratory data analysis, data visualization, and clustering. Many existing manifold-learning approaches to tensor data proceed mode by mode, often by matricizing the tensor along one mode, i.e., reordering the elements of the tensor into a matrix, and then applying a single-mode embedding method. This strategy can fail to capture important relationships between the geometries of the different modes. Tensor decompositions and tensor-network methods do couple modes, but typically through multilinear factors or bond dimensions (Acar and Yener, 2009; Kolda and Bader, 2009; Anandkumar et al., 2014; Cichocki et al., 2015; Sidiropoulos et al., 2017). Our approach addresses a complementary setting: for each mode it constructs multiscale distances between tensor slices, in order to learn nonlinear geometric embeddings.

Suppose we seek to perform dimension reduction on the rows *and* columns of a data matrix, a 2-tensor. Manifold learning techniques often focus on the observations (columns) which are measurements in a high-dimensional feature space (rows). These exploit correlations among the features to reduce the dimension of the feature vectors and recover the underlying low-dimensional geometry of the observations. Yet correlations exist among observations and features for many data matrices, e.g., data from gene expression studies, recommendation systems, sensor networks, and word-document analysis (Kalofolias et al., 2014; Bellazzi et al., 2021; Kusner et al., 2015). Exploiting correlations among both the rows and columns of a data matrix can lead to better embeddings.

Motivated by the above intuition, we present tensor co-manifold learning (TCML) – a new framework for simultaneous dimension reduction on all modes of a tensor. TCML generalizes co-manifold learning methods for matrices (Mishne et al., 2019) to higher-order tensors. TCML accounts for correlations across all modes through mode-wise multiscale metrics that are computed across multiple estimates of the tensor smoothed at different levels of granularity along each mode. We calculate a mode's multiscale metric in two steps. First, we compute smooth estimates of the tensor using a continuous relaxation of co-clustering. The level of smoothing along each mode depends on a nonnegative hyper-parameter. Smoothness along a mode monotonically increases with the hyper-parameter for that mode. Thus, different combinations of hyper-parameter values produce estimates of the tensor smoothed to different degrees along each mode. Second, we define a new multiscale metric between a pair of tensor slices in one mode that accounts for couplings across all modes by taking a weighted average of the distance between pairs of slices in each of the smoothed tensors from the first step. Once we have a multiscale metric for each mode, we use kernel-based embedding techniques to perform dimension reduction.

We make the following contributions in this paper:

- The TCML framework yields a collection of smoothed estimates of the tensor at different scales, capturing the underlying geometry of the data at multiple levels of granularity. Using these estimates, we derive a new multiscale dissimilarity for pairwise tensor slices. This dissimilarity is then used for nonlinear manifold learning to visualize and/or embed each mode of the tensor.

- We extend the co-clustering objective that TCML solves to obtain smoothed tensor estimates to recover mode-wise geometry in the missing data setting. Furthermore, we develop a compression scheme for the objective in order to speed up the solution when solving for a sequence of increasing smoothing scales.

- The dominant cost in TCML is estimating a collection of smooth tensors via co-clustering. We propose an ADMM algorithm for tensor co-clustering, which is of independent interest. The edge-wise and multiplier updates are linear in the number of edge-indexed variables, and each matrix-vector multiplication in the primal variable update costs $O(N + M_c)$, where $N = \prod_d n_d$ and $M_c = \sum_d |E_d| n_{-d}$. Thus, for sparse mode graphs, the structured linear algebra needed by the ADMM solver scales nearly linearly per matrix-vector product.

- We demonstrate the effectiveness of TCML on a variety of synthetic and real-world datasets (dense and sparse tensors). TCML outperforms existing methods in terms of both visualization and clustering performance. Specifically, TCML recovers low-dimensional coupled structure that is missed by methods that treat each mode of the tensor independently.

The organization of the paper is as follows: Section 2 introduces notation and preliminaries. Section 3 formulates the graph-regularized tensor smoothing problem that serves as the computational core of TCML, and presents the sparse-graph ADMM solver used to compute multiscale smooth tensor estimates. Section 4 introduces the full TCML framework: the framework constructs coupled multiscale dissimilarities for each tensor mode from the smoothed estimates and then uses these dissimilarities for nonlinear embedding, clustering, and visualization. Section 5 evaluates TCML on synthetic and real-world tensors, including missing-data and large sparse settings. The appendices provide convergence details, missing-data and compression extensions, and the snowflake-specific terminal $\gamma$-scale used to construct the regularization grids.

## 1.1 Related work:

**Data co-organization:** Various co-organization methods assume a manifold structure along both the rows and the columns of data matrices, (Gavish and Coifman, 2012; Ankenman, 2014; Shahid et al., 2016; Mishne et al., 2019; Düsterwald et al., 2024; Lin et al., 2025). These approaches leverage the co-dependence between the rows and columns to infer underlying row and column embeddings. Mishne et al. (2016) and Yair et al. (2017) extend this strategy to tensors. While these methods produce embeddings that recover meaningful structure, they rely on heuristic constructions of partition trees that lack algorithmic convergence guarantees. TCML is most closely related to co-manifold learning which is a strategy built on a co-clustering algorithm with convergence guarantees. Co-clustering is a simple form of co-organization that assumes an underlying smooth block structure, e.g., Jegelka et al. (2009); Wu et al. (2016); Wang and Zeng (2019); Han et al. (2022). Co-manifold learning (Mishne et al., 2019) combines co-organizing the rows and columns of a data matrix and convex optimization methods for co-clustering (Chi et al., 2017). While TCML shares core ideas with the work by Mishne et al. (2019), we emphasize that generalizing the framework to tensors requires two non-trivial innovations: i) a more scalable co-clustering algorithm (Section 3.1) and ii) an efficient procedure for selecting hyper-parameters (Appendix H.4).

**Tensor decomposition:** Low-rank tensor decompositions provide a multilinear approach to dimension reduction of multiway data (Kolda and Bader, 2009). Wu et al. (2019); Cai et al. (2023); Nie et al. (2020); Guo et al. (2023); Ji and Feng (2023) propose various multiway embedding or clustering methods that rely on tensor factorizations to learn a low-rank approximation for discovering high-order relationships. Moreover, these standard low-rank tensor factorization can be refined. For example, Guo et al. (2023) utilize a tensor logarithmic Schatten-$p$ norm to obtain a more compact low-rank structure by characterizing high-order correlations among multiple views. Ji and Feng (2023) employ a tensor decomposition to obtain a tighter approximation of the tensor rank and generate consistent and complementary factors. These tensor decomposition approaches project the modes of a tensor onto low-dimensional subspaces that may not be flexible enough to represent the data reliably. TCML differs from this line of work as we are interested in nonlinear embeddings and multiresolution representations.

**Graph-based regularization:** Graph-based regularizations along modes of the tensor are proving versatile for developing robust tensor and low-rank decompositions (Nie et al., 2017; Zhang et al., 2018; Shahid et al., 2019), as well as new approaches to problems in higher order data processing such as tensor completion, data imputation, recommendation system, feature selection, anomaly detection, and co-clustering (Li et al., 2015; Sun et al., 2018; Ioannidis et al., 2019; Su et al., 2018; Xie et al., 2018; Chi et al., 2020). Generalization of these methods for matrices to methods for tensors incurs a higher computational cost than their matrix counterparts. Thus, multiway graph-regularized formulations typically combine a low-rank tensor factorization, e.g., Canonical Polyadic (CP) or Tucker decompositions, with graph-based regularization along the rows of the factor matrices (Leonardi and Van De Ville, 2013; Xia et al., 2014; Xie et al., 2018; Ioannidis et al., 2019; Li et al., 2022). Here we use a graph-regularized formulation of tensor co-clustering to approximate a data tensor at multiple scales.

## 2 Preliminaries

We follow the notation and terminology in Kolda and Bader (2009). The number of ways or modes of a tensor determine its *order*. Vectors are tensors of order one and denoted by boldface lowercase letters, e.g., $\mathbf{a}$. Matrices are tensors of order two and denoted by boldface capital letters, e.g., $\mathbf{A}$. Tensors of higher-order, namely order three and greater, we denote by boldface Euler script letters, e.g., $\mathcal{A}$. Thus, if $\mathcal{A}$ represent a $D$-way data array of size $n_1 \times n_2 \times \cdots \times n_D$, we say $\mathcal{A}$ is a tensor of order $D$. We denote scalars by lowercase letters, e.g., $a$. We denote the $i$th element of a vector $\mathbf{a}$ by $a_i$, the $ij$th element of a matrix $\mathbf{A}$ by $a_{ij}$, the $ijk$th element of a third-order tensor $\mathcal{A}$ by $a_{ijk}$, and so on. We denote the set of indices $\{1, \ldots, D\}$ by $[D]$. We use a colon to indicate all elements of a mode. Consequently, we denote the $i$th row of a matrix $\mathbf{A}$ by $\mathbf{A}_{i:}$ and the $j$th column of a matrix $\mathbf{A}$ by $\mathbf{A}_{:j}$. *Slices* are the two-dimensional subarrays of a tensor obtained by fixing all but two indices. For example, a slice of a third-order tensor $\mathcal{A}$ is denoted by $\mathcal{A}_{i::}, \mathcal{A}_{:j:}$, or $\mathcal{A}_{::k}$.

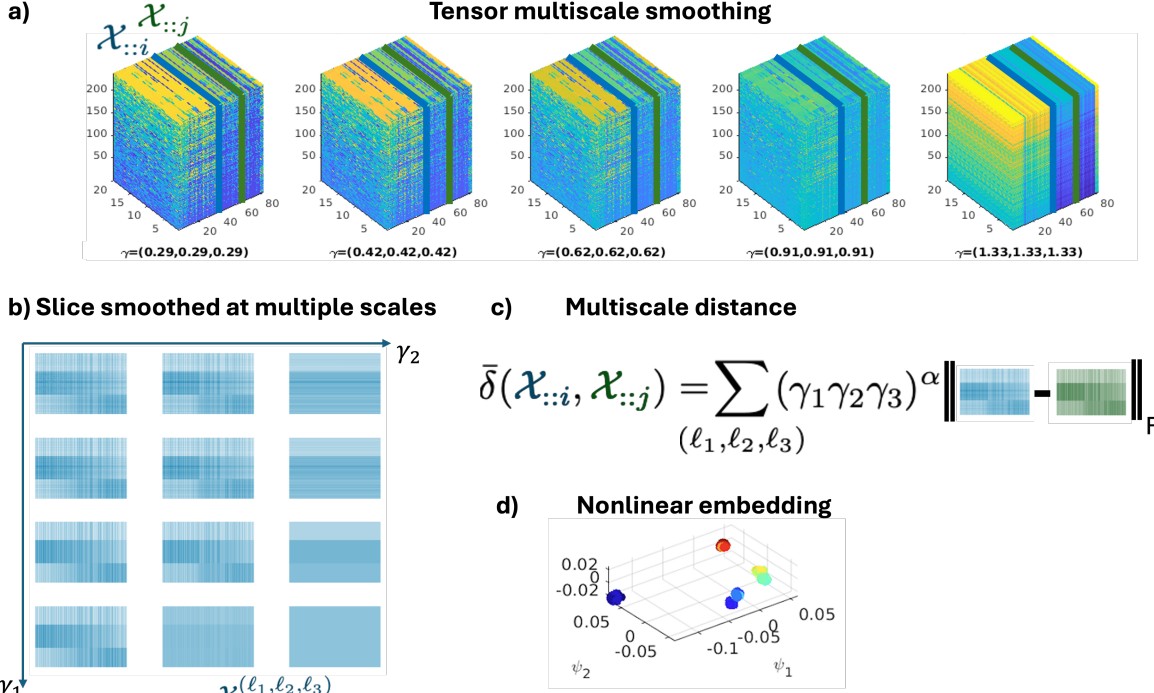

Figure 1: TCML overview. a) The input tensor is smoothed at multiple scales determined by $\{\gamma_d\}$. b) Example of a slice $\mathcal{X}_{::i}$ smoothed at different scales for increasing $\gamma_1, \gamma_2$ values. c) Multiscale distance between two slices of mode 3 $\bar{\delta}(\mathcal{X}_{::i} - \mathcal{X}_{::j})$ is a weighted sum of the Frobenius norm of the difference of the slices at multiple smoothed scales. d) Manifold learning applied to this distance yields a nonlinear embedding of the data (here visualized in 3d). Repeating this for all modes yields an embedding of each mode of the tensor.

**Basic tensor operations:**  Reordering a tensor's elements into a matrix is referred to as *matricization*, while reordering its elements into a vector is referred to as *vectorization*. In this paper, we use a canonical mode-$d$ matricization, where the mode-$d$ fibers of a $D$-way tensor $\mathcal{A} \in \mathbb{R}^{n_1 \times n_2 \times \cdots \times n_D}$ become the columns of a matrix $\mathbf{A}_{(d)} \in \mathbb{R}^{n_d \times n_{-d}}$, where $n_{-d} = \prod_{j \neq d} n_j$. In this paper, we take the vectorization of a $D$-way tensor $\mathcal{A}$, denoted $\mathrm{vec}(\mathcal{A})$, to be the column-major vectorization of the mode-1 matricization of $\mathcal{A}$, namely $\mathrm{vec}(\mathcal{A}) = \mathrm{vec}(\mathbf{A}_{(1)}) \in \mathbb{R}^n$, where $n = \prod_d n_d$ is the total number of elements in $\mathcal{A}$. As a shorthand, when the context leaves no ambiguity, we denote this vectorization of a tensor $\mathcal{A}$ by boldface lowercase $\mathbf{a}$.

The squared Frobenius norm of a $D$-way tensor $\mathcal{A} \in \mathbb{R}^{n_1 \times n_2 \times \cdots \times n_D}$ is defined as

$$\|\mathcal{A}\|_{\mathrm{F}}^2 = \sum_{i_1=1}^{n_1} \sum_{i_2=1}^{n_2} \cdots \sum_{i_D=1}^{n_D} a_{i_1 i_2 \cdots i_D}^2 = \|\mathbf{a}\|_2^2.$$

Let $\mathcal{A}$ be a tensor in $\mathbb{R}^{n_1 \times n_2 \times \cdots \times n_D}$ and $\mathbf{B}$ be a matrix in $\mathbb{R}^{m \times n_d}$. The *mode-$d$ (matrix) product* of the tensor $\mathcal{A}$ with the matrix $\mathbf{B}$, denoted by $\mathcal{A} \times_d \mathbf{B}$, is the tensor of size $n_1 \times \cdots \times n_{d-1} \times m \times n_{d+1} \times \cdots \times n_D$ whose $(i_1, i_2, \cdots, i_{d-1}, j, i_{d+1}, \cdots, i_D)$th element is given by

$$(\mathcal{A} \times_d \mathbf{B})_{i_1 \ldots i_{d-1} j i_{d+1} \cdots i_D} = \sum_{i_d=1}^{n_d} a_{i_1 i_2 \cdots i_D} b_{j i_d},$$

for $j \in [m]$. The vectorization of the mode-$d$ product $\mathcal{A} \times_d \mathbf{B}$ can be expressed as

$$\mathrm{vec}(\mathcal{A} \times_d \mathbf{B}) = (\mathbf{I}_{n_D} \otimes \cdots \otimes \mathbf{I}_{n_{d+1}} \otimes \mathbf{B} \otimes \mathbf{I}_{n_{d-1}} \otimes \cdots \otimes \mathbf{I}_{n_1})\mathbf{a}, \tag{1}$$

where $\mathbf{I}_p$ is the $p$-by-$p$ identity matrix and $\otimes$ denotes the Kronecker product between two matrices.

# 3 Smoothing a Tensor Along its Modes

We seek a collection of approximations of a data tensor $\mathcal{X} \in \mathbb{R}^{n_1 \times \cdots \times n_D}$ that have been smoothed jointly along their $D$ modes to varying degrees. We use these smooth approximations at different scales to compute a multiscale distance for each mode in Section 4.1.

Each smooth approximation is the solution to a continuous optimization problem. Specifically, we seek a minimizer $\mathcal{U}$ of the following objective function:

$$f(\mathcal{U}) \quad = \quad \frac{1}{2}\|\mathcal{X} - \mathcal{U}\|_{\mathrm{F}}^2 + \sum_{d=1}^{D} \gamma_d J_d(\mathcal{U}) \tag{2}$$

$$J_d(\mathcal{U}) \quad = \quad \sum_{e \in \mathcal{E}_d} \rho(\|\mathcal{U} \times_d \mathbf{\Delta}_{d,e}\|_{\mathrm{F}}), \quad d \in [D] \tag{3}$$

where $\mathbf{\Delta}_{d,e}$ denotes the oriented edge difference operator for edge $e$ in mode $d$. The index sets $\mathcal{E}_d$ for $d \in [D]$ denote the edge sets of mode-$d$ graphs that encode a preliminary data-driven assessment of the similarities between pairs of mode-$d$ subarrays of $\mathcal{X}$. The function $\rho$ is a nondecreasing map from $[0, \infty)$ into $[0, \infty)$.

The objective function in eq. (2) is the sum of two functions. The first term in eq. (2) quantifies how well $\mathcal{U}$ approximates $\mathcal{X}$. The sum of the roughness penalties, $J_d(\mathcal{U})$ for $d \in [D]$, in the second term incentivizes smoothness along the $D$ modes of $\mathcal{U}$, as the $d$th summand penalizes a monotonic function of the Frobenius norm of $\mathcal{U} \times_d \mathbf{\Delta}_{d,e}$ which is the difference across edge $e$ (i.e., between its incident vertices) of the mode-$d$ subarrays of $\mathcal{U}$. The nonnegative parameters $\gamma_1, \ldots, \gamma_D$ tune the tradeoff between how well $\mathcal{U}$ agrees with $\mathcal{X}$ and how smooth $\mathcal{U}$ is along its $D$ modes. By tuning $\boldsymbol{\gamma} = (\gamma_1, \ldots, \gamma_D)$, we obtain estimates of $\mathcal{X}$ at varying scales of smoothness along its $D$ modes. For example, by varying $\gamma_d$, one can obtain estimates of $\mathcal{X}$ at varying levels, or scales, of mode-$d$ smoothness (see Fig. 1a). For $\gamma_d = 0$, the tensor slices along mode $d$ are not smoothed at all. For sufficiently large $\gamma_d$ all the mode-$d$ subarrays of $\mathcal{U}$ are identical and equal the average mode-$d$ subarray of $\mathcal{X}$ (Chi et al., 2020).

The monotonic map $\rho$ enables us to differentially penalize the magnitude of $\mathcal{U} \times_d \mathbf{\Delta}_{d,e}$. We want to penalize more strongly small magnitudes, which are likely due to noise, and penalize less strongly large magnitudes, which are likely due to genuine differences between the mode-$d$ subarrays of $\mathcal{X}$ connected by edge $e$. Consequently, we focus on $\rho$ that satisfy the following criteria.

**Assumption 3.1.** The map $\rho : [0, \infty) \mapsto [0, \infty)$ is (i) concave and continuously differentiable on $(0, \infty)$, (ii) vanishes at the origin: $\rho(0) = 0$, (iii) is increasing on $[0, \infty)$, and (iv) has finite right directional derivative at the origin.

For the rest of the paper, we use the "snowflake" penalty function for $\rho$. Let $\epsilon$ be a small positive number. Then

$$\rho(z) = \frac{1}{2} \int_0^z \frac{1}{\sqrt{\zeta} + \epsilon} d\zeta. \tag{4}$$

It is straightforward to verify that the snowflake penalty in eq. (4) satisfies Assumption 3.1. The parameter $\epsilon > 0$ regularizes the right derivative at the origin. Smaller $\epsilon$ increases $\rho'(0+)$, strengthening the fusion-inducing effect on small edge differences, while larger $\epsilon$ makes the penalty closer to a smooth concave shrinkage. We treat $\epsilon$ as a fixed hyperparameter. Note that the same $\epsilon$ appears in the proximal update in Proposition 3.2 and in the terminal $\gamma$-scale derived in Appendix H.3.

Although $\rho$ is smooth as a scalar function on $(0, \infty)$, the group penalty $\mathcal{V} \mapsto \rho(\|\mathcal{V}\|_{\mathrm{F}})$ is nonsmooth at $\mathcal{V} = 0$, which incentivizes sparsity in pairwise differences. Consequently, small noisy variations between a pair of mode-$d$ subarrays are eliminated completely for sufficiently large $\gamma_d$. Since $\rho$ is concave, large pairwise differences are shrunk towards zero less. Consequently, large differences between a pair of mode-$d$ subarrays are left more intact. Other folded-concave penalties could likely work in lieu of the snowflake, e.g., logarithmic penalty, MCP, or SCAD Du (2025), to achieve the desired amount of smoothing, but we leave this as future work.

### 3.1 Splitting methods for smoothing

We use an *Alternating Direction Method of Multipliers* (ADMM) algorithm (Boyd et al., 2011) to minimize the tensor smoothing objective in eq. (2). Minimizing the objective function eq. (2) is challenging due to the concavity of $\rho$. The optimization problem is generally nonconvex because $\rho$ is concave, and it is nonsmooth because $U \mapsto \rho(\|U \times_d \Delta_{d,e}\|_F)$ is nonsmooth at zero. As mentioned previously, this nonsmoothness is intentional: it induces exact fusion of small edge differences and thereby produces co-clustered, smoothed tensor estimates. In prior work for the matrix case, Mishne et al. (2019) used a Majorization-Minimization algorithm. This strategy, however, scales prohibitively poorly to higher-order tensors (see Tab. 2). Our ADMM algorithm exploits special structure in the co-clustering problem, i.e., sparse mode graphs $\{\mathcal{E}_d\}_{d\in[D]}$, making multiscale tensor smoothing computationally tractable. In particular, the ADMM splitting introduces auxiliary variables $V_{d,e} = U \times_d \Delta_{d,e}$. This separates the nonsmooth folded-concave penalty into independent edge-wise proximal updates, while the $U$-update becomes a structured symmetric positive-definite linear system involving a Kronecker sum of sparse graph Laplacians.

For each mode $d \in [D]$ and edge $e \in \mathcal{E}_d$, define $\mathcal{V}_{d,e} = \mathcal{U} \times_d \boldsymbol{\Delta}_{d,e}$. Let $\lambda_{d,e}$ denote a Lagrange multiplier corresponding to the constraint $\mathcal{V}_{d,e} = \mathcal{U} \times_d \boldsymbol{\Delta}_{d,e}$. For $\beta > 0$, the augmented Lagrangian, $\mathcal{L}(\mathcal{U}, \mathcal{V}, \lambda)$ is

$$\frac{1}{2}\|\mathcal{X} - \mathcal{U}\|_{\mathrm{F}}^2 + \sum_{d=1}^{D} \sum_{e \in \mathcal{E}_d} \left[ \gamma_d \rho(\|\mathcal{V}_{d,e}\|_{\mathrm{F}}) + \langle \lambda_{d,e}, \mathcal{V}_{d,e} - \mathcal{U} \times_d \boldsymbol{\Delta}_{d,e} \rangle + \frac{\beta}{2}\|\mathcal{V}_{d,e} - \mathcal{U} \times_d \boldsymbol{\Delta}_{d,e}\|_{\mathrm{F}}^2 \right].$$

Our ADMM algorithm, summarized in Algorithm 1, finds stationary / KKT points of $\mathcal{L}(\mathcal{U}, \mathcal{V}, \lambda)$ by iteratively updating $\mathcal{V}, \mathcal{U}$, and $\lambda$ as follows.

**Update $\mathcal{V}$:** At iteration $k$, fix $\mathcal{U}^{(k)}$ and $\lambda^{(k)}$. For each mode $d$ and edge $e$, we solve

$$\mathcal{V}_{d,e}^{(k+1)} = \arg\min_{\mathcal{V}} \frac{\gamma_d}{\beta} \rho(\|\mathcal{V}\|_{\mathrm{F}}) + \frac{1}{2}\left\|\mathcal{V} - \left(\mathcal{U}^{(k)} \times_d \boldsymbol{\Delta}_{d,e} - \tfrac{1}{\beta}\lambda_{d,e}^{(k)}\right)\right\|_{\mathrm{F}}^2.$$

The $\mathcal{V}_{d,e}$-update requires computing the *proximal mapping* of the function $h(\mathcal{V}) = \frac{\gamma_d}{\beta}\rho(\|\mathcal{V}\|_{\mathrm{F}})$. Recall that the *proximal mapping* of a proper and lower-semicontinuous function $h$ is

$$\mathrm{prox}_h(\mathbf{z}) = \arg\min_{\mathbf{v}} \frac{1}{2}\|\mathbf{z} - \mathbf{v}\|_2^2 + h(\mathbf{v}).$$

The function $h(\mathcal{V})$ is proper, lower-semicontinuous, and has an analytical expression for its proximal map.

**Proposition 3.2.** *Let $\rho$ denote the snowflake penalty function eq. (4), and let $x := \|\mathcal{V}\|_{\mathrm{F}}$. Then*

$$\mathrm{prox}_{\upsilon\rho(\|\cdot\|_F)}(\mathcal{V}) = \frac{\phi_*^2}{\|\mathcal{V}\|_F}\mathcal{V},$$

*where $\phi_* = \arg\min_{\phi \in C} \left[\upsilon\rho(\phi^2) + \frac{1}{2}(\phi^2 - x)^2\right]$ and*

$$C = \left\{\phi : \phi \geq 0, \ \phi\left[\phi^3 + \epsilon\phi^2 - x(\phi + \epsilon) + \frac{\upsilon}{2}\right] = 0\right\}.$$

The proof is in the Supplement. Note that the set $C$ has at most four elements which can be enumerated using Cardano's formula.

**Update $\mathcal{U}$:** Next, we fix $\mathcal{V}^{(k+1)}, \lambda^{(k)}$ and solve

$$\min_{\mathcal{U}} \frac{1}{2}\|\mathcal{X} - \mathcal{U}\|_{\mathrm{F}}^2 + \sum_{d=1}^{D} \sum_{e \in \mathcal{E}_d} \frac{\beta}{2}\left\|\mathcal{V}_{d,e}^{(k+1)} - \mathcal{U} \times_d \boldsymbol{\Delta}_{d,e} + \tfrac{1}{\beta}\lambda_{d,e}^{(k)}\right\|_{\mathrm{F}}^2,$$

which requires solving the linear system:

$$\left(\mathbf{I} + \beta \sum_{d=1}^{D} \mathcal{L}^{(d)}\right)\mathrm{vec}(\mathcal{U}^{(k+1)}) = \mathrm{vec}(\mathcal{X}) + \sum_{d=1}^{D} \mathbf{A}_d^{\mathsf{T}}\left[\beta\mathrm{vec}(\mathcal{V}_d^{(k+1)}) + \mathrm{vec}(\lambda_d^{(k)})\right], \tag{5}$$

where each $\mathcal{L}^{(d)}$ is a Laplacian operator associated with $\boldsymbol{\Delta}_{d,e}$, $\mathbf{A}_d = \mathbf{I}_{n_D} \otimes \cdots \otimes \mathbf{I}_{n_{d+1}} \otimes \boldsymbol{\Phi}_d \otimes \mathbf{I}_{n_{d-1}} \otimes \cdots \otimes \mathbf{I}_{n_1}$, and $\boldsymbol{\Phi}_d$ is the oriented edge-vertex incidence matrix for the mode-$d$ graph, i.e., the $l$th row of $\boldsymbol{\Phi}_d$ is $\boldsymbol{\Delta}_{d,l}$.

**Update** $\lambda$**:** Finally, we update each multiplier: $\lambda_{d,e}^{(k+1)} = \lambda_{d,e}^{(k)} + \beta\left[\mathcal{V}_{d,e}^{(k+1)} - \mathcal{U}^{(k+1)} \times_d \boldsymbol{\Delta}_{d,e}\right]$.

**Sparsity and per-iteration cost.** Let $N := \prod_{d=1}^{D} n_d$ denote the total number of entries in the data tensor $\mathcal{X}$, and let $n_{-d} := \prod_{j \neq d} n_j = N/n_d$ for each mode $d$. Also write $m_d := |\mathcal{E}_d|$ for the number of edges in the mode-$d$ similarity graph. The data tensor $\mathcal{X}$ and the primal variable $\mathcal{U}$ each contain $N$ entries.

For a fixed edge $e \in \mathcal{E}_d$, the auxiliary block $\mathcal{V}_{d,e} = \mathcal{U} \times_d \boldsymbol{\Delta}_{d,e}$ is a $(D-1)$-way tensor with $n_{-d}$ entries. The same is true of $\mathcal{Z}_{d,e}$ and $\lambda_{d,e}$. Hence the edge-indexed variables for mode $d$ require $O(m_d n_{-d})$ storage, and the total auxiliary storage is $\widehat{M} := \sum_{d=1}^{D} m_d n_{-d}$, and if the mode graphs are sparse, i.e. $m_d = O(n_d)$, then $\widehat{M} = \sum_{d=1}^{D} O(n_d n_{-d}) = O(DN)$. The $\mathcal{V}$- and $\lambda$-updates are block / coordinate-wise operations on these edge variables. Computing $\mathcal{Z}_{d,e}^{(k)} = \mathcal{U}^{(k)} \times_d \boldsymbol{\Delta}_{d,e} - \frac{1}{\beta}\lambda_{d,e}^{(k)}$, evaluating the proximal map for $\mathcal{V}_{d,e}^{(k+1)}$, and updating $\lambda_{d,e}^{(k+1)}$ each cost $O(n_{-d})$ for one edge. Summing over all modes and edges, these two updates therefore cost $O(\widehat{M})$ per ADMM iteration.

The most expensive calculation is the $\mathcal{U}$-update in the system in eq. (5), whose coefficient matrix is a Kronecker-sum operator that need not be formed explicitly. Instead, one applies it implicitly by reshaping $\mathrm{vec}(\mathcal{U})$ into its mode-$d$ matricization, multiplying by the sparse mode-$d$ Laplacian, and re-vectorizing. Since the system in eq. (5) has $O(m_d)$ nonzeros, applying the $d$th term costs $O(m_d n_{-d})$, so one matrix-vector multiply with the full left-hand side costs $O\left(N + \widehat{M}\right)$. Thus, for fixed tensor order $D$, each ADMM iteration is nearly linear in the number of tensor entries, up to the cost of solving the symmetric positive definite system in eq. (5). In particular, using a nearly-linear-time solver such as Spielman and Teng (2004) yields an $\widetilde{O}(N + \widehat{M})$ $\mathcal{U}$-update, although in practice we use the Preconditioned Conjugate Gradient method, which is empirically much faster.

Algorithm 1 comes with the following convergence guarantees. The proof is in the appendix.

**Theorem 3.3.** *For sufficiently large* $\beta$, *the limit points of the sequence of iterates* $\{(\mathcal{U}^k, \mathcal{V}^k, \lambda^k)\}_{k \geq 0}$ *of Algorithm 1 are stationary / KKT points of* $\mathcal{L}(\mathcal{U}, \mathcal{V}, \lambda)$.

### 3.2 Practical considerations and extensions

In this section, we describe two practical extensions of the multiscale tensor smoothing algorithm used by TCML: smoothing in the missing-data setting and compression for large-scale tensors.

---

**Algorithm 1** ADMM for Tensor Smoothing

**Require:** $\mathcal{X} \in \mathbb{R}^{n_1 \times \cdots \times n_D}$, graphs $\{\mathcal{E}_d\}_{d=1}^{D}$, penalty parameters $\{\gamma_d\}$, snowflake parameter $\epsilon > 0$, ADMM parameter $\beta > 0$.

1: Initialize $\mathcal{U}^{(0)}, \mathcal{V}_{d,e}^{(0)}, \lambda_{d,e}^{(0)}$ for all $e \in \mathcal{E}_d$.
2: **for** $k = 0, 1, \dots$ until convergence **do**
3:   $\mathcal{Z}_{d,e}^{(k)} \leftarrow \mathcal{U}^{(k)} \times_d \boldsymbol{\Delta}_{d,e} - \frac{1}{\beta}\lambda_{d,e}^{(k)}$   $\forall e \in \mathcal{E}_d$.
4:   $\mathcal{V}_{d,e}^{(k+1)} \leftarrow \mathrm{prox}_{\frac{\gamma_d}{\beta}\rho}\left(\mathcal{Z}_{d,e}^{(k)}\right)$   $\forall e \in \mathcal{E}_d$.
5:   Solve eq. (5) to obtain $\mathcal{U}^{(k+1)}$
6:   $\lambda_{d,e}^{(k+1)} \leftarrow \lambda_{d,e}^{(k)} + \beta\left[\mathcal{V}_{d,e}^{(k+1)} - \mathcal{U}^{(k+1)} \times_d \boldsymbol{\Delta}_{d,e}\right]$.
7:   Check convergence (e.g. $\|\mathcal{U}^{(k+1)} - \mathcal{U}^{(k)}\|_{\mathrm{F}} \leq \epsilon$).
8: **end for**

---

**Missing data.** Suppose $\mathcal{X} \in \mathbb{R}^{n_1 \times \cdots \times n_D}$ is observed only over entries $\Theta \subseteq \prod_{d=1}^{D}[n_d]$. The index set of unobserved entries is $\Theta^c$. We extend TCML to the missing data scenario by solving the multi-way smoothing problem over the observed entries, i.e., we replace the fidelity term in eq. (2) with $\frac{1}{2}\|P_\Theta(\mathcal{X}) - P_\Theta(\mathcal{U})\|_{\mathrm{F}}^2$, where $\mathcal{P}_\Theta$ denotes the projection onto the observed indices. Running TCML over a set of scale parameters $\{\gamma_d\}$ yields a scale–indexed family of smooth tensors $\{\mathcal{U}^{(\ell_1, \dots, \ell_D)}\}_{\ell_d}$. Each such tensor is then used to impute the missing values via $\widetilde{\mathcal{X}}^{(\ell_1, \dots, \ell_D)} = \mathcal{P}_\Theta(\mathcal{X}) + \mathcal{P}_{\Theta^c}(\mathcal{U}^{(\ell_1, \dots, \ell_D)})$, thus producing a multiscale collection of imputed tensors. The full details are described in the Supplement.

**Compression.** The ADMM solution for a set of scale parameters $\{\gamma_d\}$ identifies clusters along each mode. To accelerate the ADMM iterations, we generalize the compression method of Yi et al. (2021) to solve smaller problems at subsequent scales. Each mode $d$ is partitioned into $k_d \ll n_d$ clusters $c_d[i] \subseteq [n_d]$; the weighted

block average yields a compressed tensor $\widetilde{\mathcal{X}} \in \mathbb{R}^{k_1 \times \cdots \times k_D}$. We then apply TCML to the smaller problem

$$\min_{\mathcal{U} \in \mathbb{R}^{k_1 \times \cdots \times k_D}} \frac{1}{2} \|\widehat{\mathcal{W}} \odot (\widetilde{\mathcal{X}} - \mathcal{U})\|_\mathrm{F}^2 + \gamma \sum_{d=1}^D \sum_{e \in \widehat{\mathcal{E}}_d} \widehat{w}_{d,e}\, \rho(\|\mathcal{U} \times_d \widehat{\boldsymbol{\Delta}}_{d,e}\|_F),$$

where $\widehat{\mathcal{W}}$ weights the difference in each element in the compressed tensor with respect to the size of the block in the original tensor so the fidelity term is equivalent (full details in Appendix F). Because each entry of $\mathcal{U}$ represents a block-constant value over its parent cluster, increasing $\gamma$ coarsens existing clusters, so the same compressed formulation is valid along the entire regularization path. This strategy shrinks the number of decision variables in the ADMM sub-problems by a factor of $\prod_d(n_d/k_d)$ while preserving the multiscale structure, leading to large runtime savings without sacrificing performance (see A.3).

## 4    Co-manifold learning on tensors

Graph-based manifold learning, e.g., Isomap (Tenenbaum et al., 2000), Laplacian eigenmaps (Belkin and Niyogi, 2003), Diffusion maps (Coifman and Lafon, 2006), t-SNE (Van der Maaten and Hinton, 2008), and UMAP (McInnes et al., 2018), relies on the construction of a similarity measure between samples. Here we use a collection of approximations of the tensor $\mathcal{X}$ smoothed at different scales to define a new metric between pairs of slices for each mode of the tensor. We convert the smoothed tensors $\{\mathcal{U}\}$ into multiscale, mode-wise distances, and subsequently into a similarity kernel for manifold embedding. This introduces a coupling between the tensor modes when embedding.

### 4.1    Multiscale metric and embedding

**Smoothing at multiple scales.** For each mode $d$, we have a regularization parameter $\gamma_d$. Solving the *multiway* tensor smoothing problem for the set of parameters $\{\gamma_d\} = (\gamma_{\ell_1}, \gamma_{\ell_2}, \cdots, \gamma_{\ell_D})$ yields a smoothed tensor $\mathcal{U}^{(\ell_1, \ldots, \ell_D)}$, where $\ell_d$ indexes the "scale" used for mode $d$. Equivalently, $\ell_d$ can be used to track a sequence of $\gamma_d$-values along mode $d$. More details on selecting $\{\gamma_d\}$ are provided in Supplement H.4.

**Multiscale distance.** For notational clarity, we consider $D = 3$ and without loss of generality let $d = 1$, to define a distance between two slices in the first mode of a tensor of order 3, however we note this definition generalizes to other tensor slices and $D > 3$. Let $\mathcal{X}_{i::}^{(\ell_1, \ell_2, \ell_3)}$ and $\mathcal{X}_{j::}^{(\ell_1, \ell_2, \ell_3)}$ denote the $i$th and $j$th slices of mode $d = 1$ at scale $(\ell_1, \ell_2, \ell_3)$.

In the generic case of embedding a mode of a tensor, e.g., of mode $d = 1$, one typically defines the distance between slices as the Frobenius norm of the difference between slices to be $\delta_\mathrm{F}\big(\mathcal{X}_{i::},\, \mathcal{X}_{j::}\big) = \big\|\mathcal{X}_{i::} - \mathcal{X}_{j::}\big\|_\mathrm{F}$. In contrast, we define the distance between the two slices using the sequence of smoothing parameters and estimates. We sample a set of parameter tuples $\big\{(\gamma_1, \gamma_2, \gamma_3)\big\}$ or equivalently, an index set $\{(\ell_1, \ell_2, \ell_3)\}$, and solve for each $\mathcal{U}^{(\ell_1, \ell_2, \ell_3)}$. At a single scale $(\ell_1, \ell_2, \ell_3)$ we define the weighted distance:

$$\delta\big(\mathcal{X}_{i::}^{(\ell_1, \ell_2, \ell_3)},\, \mathcal{X}_{j::}^{(\ell_1, \ell_2, \ell_3)}\big) = \big(\gamma_{\ell_1}\gamma_{\ell_2}\gamma_{\ell_3}\big)^\alpha \big\|\mathcal{U}_{i::}^{(\ell_1, \ell_2, \ell_3)} - \mathcal{U}_{j::}^{(\ell_1, \ell_2, \ell_3)}\big\|_\mathrm{F}. \tag{6}$$

The factor $(\gamma_{\ell_1}\gamma_{\ell_2}\gamma_{\ell_3})^\alpha$ weights how much each scale of smoothing contributes to the overall distance. We set $\alpha < 0$ if we want to emphasize "local" scales over globally merged ones or $\alpha > 0$ to emphasize global structure. We then *sum* the scale-weighted distances to obtain the final multiscale distance between the $i$th and $j$th subarrays along mode $d = 1$ (Fig. 1c):

$$\bar{\delta}\big(\mathcal{X}_{i::}, \mathcal{X}_{j::}\big) = \sum_{(\ell_1, \ell_2, \ell_3)} \delta\big(\mathcal{X}_{i::}^{(\ell_1, \ell_2, \ell_3)},\, \mathcal{X}_{j::}^{(\ell_1, \ell_2, \ell_3)}\big), \tag{7}$$

where the sum ranges over all sampled scale tuples. Note that

$$\bar{\delta}\big(\mathcal{X}_{i::}, \mathcal{X}_{j::}\big) = \bar{\delta}\big(\mathcal{X}_{j::}, \mathcal{X}_{i::}\big) \geq 0. \tag{8}$$

This construction incorporates both locally (small $\gamma_d$) and globally (large $\gamma_d$) smoothed versions of the data in each mode.

The distances in eq. (6)–eq. (7) yield a family of *multiscale similarity measures* on the slices of $\mathcal{X}$ via multiway smoothing across modes. In Supplement E, we present an adaptation of these similarity measures to the missing data setting. Briefly, TCML solutions impute the missing values with increasingly smoother estimates. This alleviates the need to identify a single scale at which to impute the entries in such a setting.

**Manifold embedding.** Let $\mathcal{X}_{d,i} = \mathcal{X} \times_d \mathbf{e}_i$ denote the $i$th subarray along mode-$d$. Then, given a multiscale distance $\overline{\delta}(\mathcal{X}_{d,i}, \mathcal{X}_{d,j})$, we form a kernel between slices of the tensor along mode $d$

$$\mathbf{K}_d(i,j) = \exp\left\{-\frac{\overline{\delta}(\mathcal{X}_{d,i}, \mathcal{X}_{d,j})^2}{\sigma^2}\right\}, \tag{9}$$

where $\sigma$ is a scale parameter, e.g., the median of all pairwise distances. This similarity kernel can then be used in graph-based manifold learning techniques (Fig. 1d). For example, using this kernel as input to Diffusion maps yields a global low-dimensional embedding that preserves the local structure, is equipped with a meaningful Euclidean distance in the embedding space, and is robust to noise. The similarity kernel can also be an input to neighbor embeddings, e.g., t-SNE or UMAP. Thus, our approach enables visualization of the data and provides a meaningful representation to recover clusters or continuous latents, while accounting for coupling between the modes of the tensor.

## 5 Numerical Experiments

We evaluate TCML on a variety of simulated and real data examples. One key choice is the selection of the $\beta$ parameter. In the Supplement, we describe the adaptive strategy used to select the ADMM parameters. For all experiments, we use the default parameters suggested in Boyd et al. (2011); with details in the supplement.

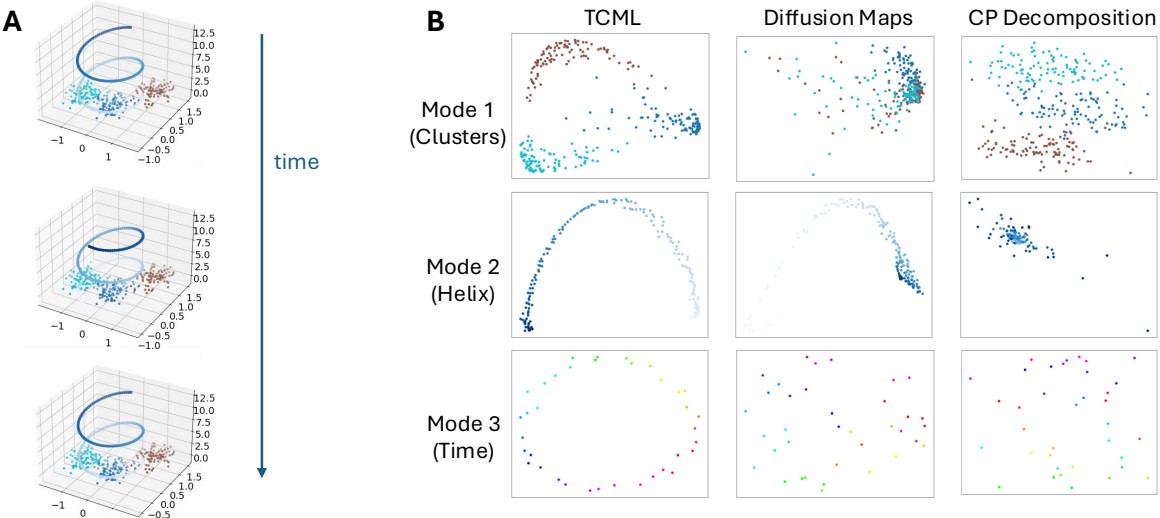

Figure 2: Time-varying linkage2. A tensor of order 3 constructed by stacking pairwise distance matrices between points on a rotating helix (1D manifold) and samples drawn from three Gaussians distributions (clustered structure). A) The helix and Gaussian clusters (the latent factors in the first two modes of the tensor) at three separate time points as the helix revolves. B) We compare TCML under a missing data setting, where 20 % of the entries in the tensor have been masked, to Diffusion maps applied independently to each mode of the tensor and CP decomposition applied to the tensor. Each row presents the recovered components for that mode of the tensor.

## 5.1 Synthetic data

**Time-varying linkage1:** We introduce a synthetic time-varying linkage dataset designed to facilitate the exploration of co-manifold learning methodologies on tensor-structured data. The dataset is generated from two distinct manifolds—a one-dimensional (1D) helix and a two-dimensional (2D) surface—both embedded in a three-dimensional (3D) Euclidean space (See Fig. 4A). Details are provided in the Supplement.

**Time-varying linkage2:** The time-varying linkage2 dataset is generated from two distinct manifolds—samples from a rotating 1D helix and from a mixture of clusters, i.e., points $\{y_j\}_{j=1}^{n_c} \in \mathbb{R}^3$ sampled from three Gaussian distributions in 3D (see Fig. 2A).

**Evaluation:** We compare TCML to Diffusion Maps (DM) (Coifman and Lafon, 2006) and the CP decomposition (Carroll and Chang, 1970; Harshman, 1970). Comparing to DM demonstrates contrasts *single-mode* manifold learning on a single-scale metric to our *multi-way multiscale* approach. Comparing to CP decomposition contrasts a *multilinear* tensor approach to our *nonlinear* manifold learning approach.

In Fig. 2B we present for each method its first two components along each mode. For the first mode, which corresponds to points on the clusters, TCML recovers 3 clusters. The other methods recover clusters that are somewhat ordered along their second component but mixed together along the first. For the second mode, which corresponds to points on the helix, TCML recovers a 1D curve. DM also recovers an embedding that is primarily ordered along the helix height $\theta_i$, however the width of the embedding reflects sensitivity to a noisy representation arising due to missing data. CP struggles to recover a meaningful second mode embedding. For the third mode, time is a 1D periodic parameter. TCML correctly recovers a circular embedding. Neither DM nor CP recover a meaningful third mode embedding. Evaluation of time-varying linkage1 is in Appendix A.

Manifold learning is not only used for visualization but also for finding new representations for downstream signal processing and machine learning tasks. Here, we calculate the accuracy of clustering applied to the low-dimensional representation of the first mode of the tensor (corresponding to 3 Gaussian clusters). We apply $k$-means with $k$ set to the correct number of clusters in the data, as we want to evaluate the ability of the methods to properly represent the data without being sensitive to the empirical estimation of the number of clusters in the data. We use the Adjusted Rand Index (ARI) Hubert and Arabie (1985) to quantify agreement between the $k$-means clustering of the embedding and the ground-truth labels. ARI indicates no agreement between two clusterings by 0 and perfect agreement by 1. In Fig. 3, we plot ARI for the three methods for an increasing percentage of random missing values in the tensor (from 10% to 90%), where we averaged over 10 realizations of missing entries. TCML (red) gives the best clustering result across all values and its performance is degraded only at 90% missing values, as opposed to DM

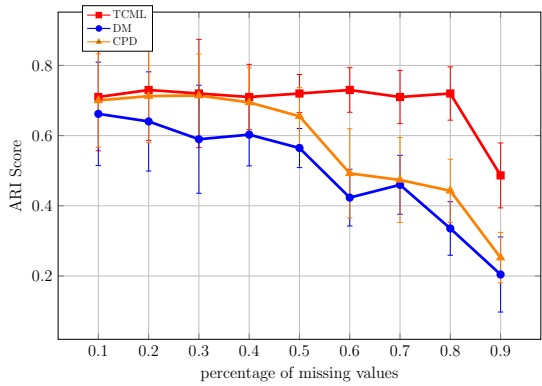

Figure 3: Time-varying linkage2. Evaluation of k-means clustering applied to embeddings of data with increasing amounts of missing values compared to the ground-truth labels of the clusters. Each plot is the average over 10 random realizations.

(blue) which performs worse. CP (orange) has similar performance to TCML for low values of missingness and degrades at 50%.

**Runtime and convergence:** We track ADMM's convergence with primal and dual residuals. For simplicity, for a given tensor slice $d$, let $H = \Delta_d$ be the incidence matrix of the mode-$d$ graph, and set

$$r_d^k = \text{vec}(V_d^k) - H\, u_d^k, \qquad d_d^k = \beta\, H^\top (u_d^k - u_d^{k-1}).$$

The residual $r_d^k$ vanishes as the iterates approach feasibility, and $d_d^k$ vanishes as the dual variables converge.

The top row of Tab. 1 reports the sum of the norms of both residuals at 6 different iteration counts. In the second row, we define $C^k \in \{0,1\}^{n \times 3}$ (corresponding to the three Gaussian clusters) to be the $k$-means

Table 1: Primal+Dual residual ($\|r_d^k\|_2 + \|d_d^k\|_2$) and Cluster stability ($\|C^k - C^{k-1}\|_1$). Standard error over 10 trials is reported in parenthesis.

| $k$ | 10 | 100 | 200 | 400 | 800 | 1000 |
|---|---|---|---|---|---|---|
| $\|r_d^k\|_2 + \|d_d^k\|_2$ | 16.7 (0.8) | 10.2 (0.6) | 5.6 (0.4) | 3.1 (0.3) | 1.7 (0.2) | 0.93 (0.10) |
| $\|C^k - C^{k-1}\|_1$ | 16 (2.1) | 11 (1.8) | 2 (0.7) | 0 (0.0) | 0 (0.0) | 0 (0.0) |

cluster indicator matrix; i.e., $C_{ij} \in \{0, 1\}$ indicates the assignment of the $i$-th sample to the $j$-th cluster. We measure cluster stability as the disagreement between successive cluster indicator matrices via the $\ell_1$-norm difference. We see rapid convergence of ADMM with respect to the primal and dual residuals. Although the residuals are still largely nonzero, we emphasize rapid initial convergence and that the cluster assignments correspondingly stabilize.

Table 2: Noise rates and ARI / Runtime (s)

**Noise rate = 0.0**

| # Samples | ADMM (ours) | + Compr. (ours) | MM |
|---|---|---|---|
| 1e3 | 0.78 (1.4) / 0.98 (0.8) | 0.78 (1.12) / 0.98 (0.8) | 0.78 (1.1) / 37.4 (12.9) |
| 1e4 | 0.79 (1.1) / 11.49 (1.9) | 0.79 (1.07) / 7.43 (0.8) | 0.79 (1.2) / 214.31 (31.4) |
| 1e5 | 0.81 (1.9) / 148.31 (7.4) | 0.81 (1.07) / 56.4 (2.8) | 0.81 (1.1) / 1227.8 (98.2) |
| 1e6 | 0.81 (1.9) / 1825.0 (29.1) | 0.81 (1.07) / 428.0 (21.4) | 0.81 (1.1) / 7032.0 (351.6) |

**Noise rate = 0.5**

| Sample rate | ADMM (ours) | + Compr. (ours) | MM |
|---|---|---|---|
| 1e3 | 0.70(1.5)/1.08(0.9) | 0.70 (1.3) /1.08 (0.9) | 0.70 (1.3) / 41.14 (14.3) |
| 1e4 | 0.71 (1.4) / 12.64 (2.1) | 0.71 (1.2) / 8.17 (0.9) | 0.71 (1.2) / 235.74 (34.6) |
| 1e5 | 0.73 (1.3) / 163.14 (8.2) | 0.73 (1.3) / 62.04 (3.1) | 0.73 (1.3) / 1350.58 (108.0) |
| 1e6 | 0.73 (1.3) / 2007.50 (32.0) | 0.73 (1.3) / 470.80 (23.4) | 0.73 (1.3) / 7735.20 (386.8) |

In Tab. 2, we provide a runtime comparison to demonstrate faster per-iteration solves compared to the Majorization-Minimization (MM) algorithm developed in Mishne et al. (2019). In this experiment, we generate a point-cloud composed of three Gaussian clusters. For various noise rates (percent missing values) and sample rates (number of data points), we evaluate the runtime and ARI score of cluster assignments derived from our method in the specific single-way case.

## 5.2 Real tensor data

**YaleFaces:** We compare TCML to two tensor decomposition-based methods on the Yale Faces Dataset Yale (2001), a 3-D tensor with axis corresponding to (individual, frames, pixels) comparing our method to the Tucker decomposition. Visualizations are provided in the Supplement in Fig. 5. All methods yield a roughly 2D embedding with the structure of an annulus, although the Tucker decomposition collapses under gross pixel masking. The embedded point clouds are observed to be overlapping rings. Each ring corresponds to an individual. Naturally, the rings correlate with the angle of the light source. In Tab. 3, we measure three qualitative statistics about the embedding: (1.) trustworthiness (van der Maaten, 2009) (a measure of neighborhood preservation) and (2.) the correlation between the angle of a sample from a reference vector and the mean pixel intensity of the image (3.) silhouette score (Rousseeuw, 1987) of the embedding and the classes identifying individuals (the overlap between the clustering and a set of labels).

Both TCML and Tucker exhibit robustness under gross pixel masking and both embeddings appropriately capture illumination. However, TCML is significantly better at clustering individuals, which we attribute to the nonconvex graph-based attraction mechanism.

**NeurIPS publication tensor** In Fig. 6 in the supplement, we present a qualitative application of TCML to an embedding of a large-scale tensor (450M elements) comprised of words, documents, and author co-occurrences, using the NeurIPS tensor in Globerson et al. (2004). We use this experiment as a qualitative

Table 3: YaleFaces tensor under 40% and 60% masked pixels. Truthfulness, Angle-Intensity Correlation (AIC), and Individual Silhouette (SIL) of 2D embeddings for various tensor decomposition methods. The standard error over 10 trials is reported in parentheses.

| | 40% Masked Pixels | | | 60% Masked Pixels | | |
| | Truth | AIC | SIL | Truth | AIC | SIL |
|---|---|---|---|---|---|---|
| TCML | 0.95 (0.01) | 0.76 (0.02) | 0.53 (0.02) | 0.90 (0.02) | 0.70 (0.03) | 0.48 (0.03) |
| Tensor Train | 0.92 (0.01) | 0.69 (0.03) | 0.51 (0.02) | 0.86 (0.02) | 0.60 (0.04) | 0.45 (0.03) |
| Tucker | 0.91 (0.01) | 0.73 (0.03) | 0.46 (0.03) | 0.85 (0.02) | 0.65 (0.04) | 0.40 (0.04) |
| CP | 0.88 (0.02) | 0.61 (0.04) | 0.30 (0.03) | 0.80 (0.03) | 0.50 (0.05) | 0.25 (0.04) |
| Kernel CP | 0.93 (0.02) | 0.72 (0.03) | 0.47 (0.03) | 0.86 (0.03) | 0.62 (0.04) | 0.38 (0.04) |

large-scale visualization of TCML on a sparse word-document-author tensor. The labels are generated after the embedding from side information and are used only to aid interpretation of visible regions of the plot; they are not used to compute the TCML distances or embeddings.

## 6 Conclusions

We propose a new framework, TCML, to smooth a data tensor at different scales along its modes. These smoothed tensors are used to define a multiscale dissimilarity that accounts for coupled dependencies among the modes. The resulting multiscale dissimilarity is then used to learn nonlinear manifold representations of each mode. We demonstrate our TCML embeddings for visualization, clustering and interpreting complex tensor data.

In order to scale TCML to large datasets, we developed an efficient Alternating Direction Method of Multipliers (ADMM) algorithm that exploits sparse mode graphs to achieve nearly linear per-iteration complexity. We coupled ADMM with a novel multiway compression strategy to scale to tensors with hundreds of millions of elements, as we demonstrated for the NeurIPS publication tensor. Finally, our empirical evaluations on synthetic manifolds and real-world image data illustrated that TCML is robust to noise and gross missingness.

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

# A    Additional experiments

## A.1    Synthetic data

**Description of datasets:**    Data from the helix $\{z_i\}_{i=1}^{n_c}$ for time-varying linkage1 and linkage2 consists of $n_c = 190$ points uniformly sampled along the 1D curve. Points from the helix have coordinates:

$$z_i = (\sin(\theta_i), \cos(\theta_i), \theta_i), \quad \theta_i \in [0, 4\pi]. \tag{10}$$

This configuration ensures multiple turns of the helix, providing sufficient complexity for manifold learning algorithms. The helix rotates around its central axis at a constant speed, completing a full revolution over $n_t = 36$ time points. Thus, time is a third 1D manifold with periodic conditions.

For time-varying linkage1, data from the 2D surface consists of $n_r = 300$ points uniformly sampled from a rectangular grid. Points from the surface have coordinates:

$$\widetilde{Y}_{k\ell} = \left(u_k, v_\ell, u_k^2 - v_\ell^2\right), \quad u_k \in [-1, 1], \ v_\ell \in [-1, 1],$$

with 20 points along the $u$-axis and 15 points along the $v$-axis.

For every time point, we calculate the pairwise distances between a point on the helix $z_i(t)$ and a point $y_j$ on the surface $\widetilde{Y}$ (for linkage1) or in the clusters (for linkage2) to obtain a tensor $\mathcal{X} \in \mathbb{R}^{n_r \times n_c \times n_t}$ whose $ijt$th element $\mathcal{X}_{ijt} = \|z_i(t) - \widetilde{y}_j\|^2$.

**Time-varying linkage1:**    In Fig. 4B we qualitatively compare TCML to Diffusion Maps (DM) (Coifman and Lafon, 2006) and to the CP decomposition (Carroll and Chang, 1970; Harshman, 1970). For each method, we present its first two components along each mode. For the first mode, which corresponds to points on the curved plane, TCML recovers a 2D manifold of points along a slightly curved grid, whereas the other two methods fail to recover a meaningful 2D manifold. For the second mode, which corresponds to points on the helix, TCML recovers a 1D curve. DM and CP also recover an embedding that is primarily ordered along the height of the helix $\theta_i$, however the thickness of the embedding indicates a sensitivity to a noisy representation arising from the missing data. For the third mode, time is a 1D periodic parameter, which is currently recovered by TCML as a circular embedding. Neither DM nor CP recover a meaningful embedding.

The **YaleFaces** dataset is a 3-D tensor with axes corresponding to (individual, frames, pixels). All methods yield a roughly 2D embedding with the structure of an annulus, although the Tucker decomposition collapses under gross pixel masking. The embedded point clouds are observed to be overlapping rings. Each ring corresponds to an individual. Naturally, the rings correlate with the angle of the light source.

Comparing the three methods, we see that all three methods generally form the shape of a connected annulus. Note that this is not necessarily the underlying geometry of the YaleFaces data, since the azimuth of the light source does not rotate 360 degrees. However, we observe that the TCML C-shape dominates: the two ends of the "C" stay apart, while color varies smoothly. Similarly, the manifold embedding is less perturbed by noise (i.e., fewer outliers).

## A.2    NeurIPS Publications Tensor

We also present a qualitative analysis of our method applied to embedding a tensor of words, documents and authors. We used the NeurIPS word-author-document co-occurrence tensor provided in Globerson et al. (2004). We preprocess the data by considering the sub-tensor corresponding to the 2000 most frequent words, excluding the first 100, the 500 authors with highest word counts, and papers published in the last four years of the dataset (1999 - 2003). This results in a tensor with shape (431, 500, 2000) with approximately 450M elements of which 29376 are nonzero entries. We use TCML to generate a multiscale embedding of documents and authors. We then generate cluster assignments using an LLM applied to authors' names and to paper titles.

In Fig. 6 we present TCML embeddings, obtained by applying t-SNE to the multiscale metric eq. (9), where points represent either words or authors in a two-dimensional space. Each point is colored according to its

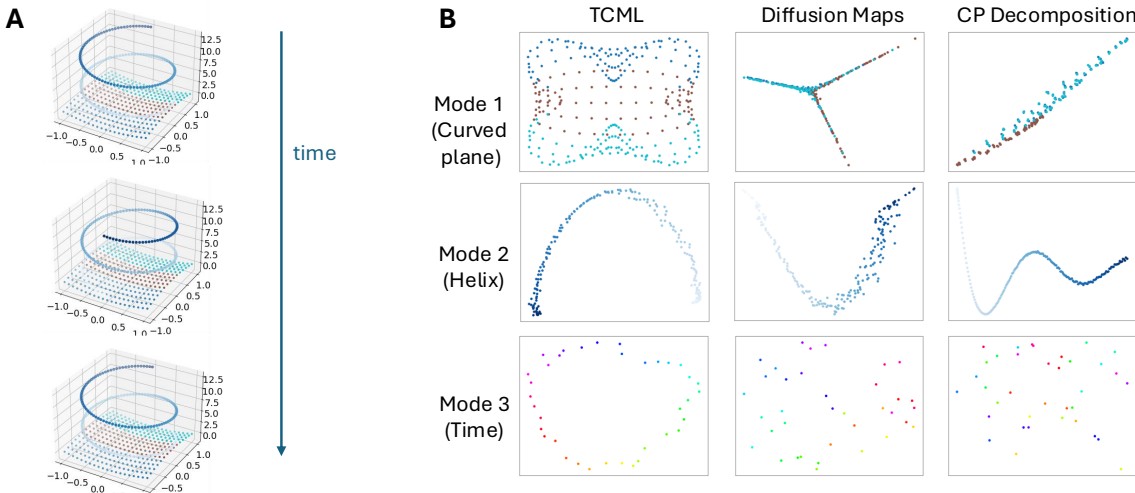

Figure 4: Time-varying linkage1. A tensor of order 3 is constructed by stacking the pairwise distance matrices between points on a rotating helix (1D manifold) and points from a curved surface (2D manifold). A) The helix and the surface (corresponding to the latent factors in the first two modes of the tensor) at three separate time points as the helix revolves. B) We compare TCML under a missing data setting, where 20 % of the entries in the tensor have been removed, to Diffusion maps applied independently to each mode of the tensor and CP decomposition applied to the tensor. Each row presented the recovered components for that mode of the tensor.

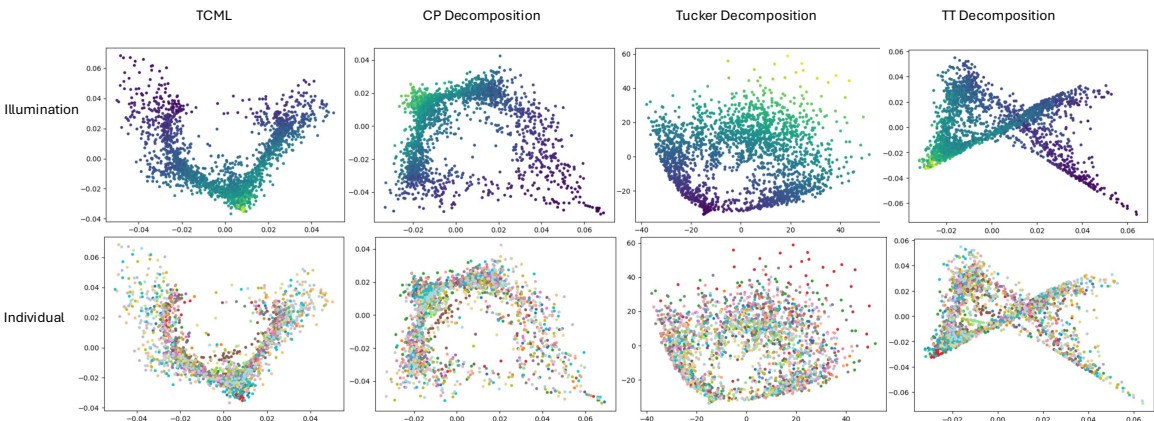

Figure 5: Yalefaces. Each point is one grayscale face image of a subject captured at a fixed pose under a wide range of lighting directions/intensities). Shown are 2D embeddings of the same dataset produced by three methods (columns): TCML, CP decomposition, and Tucker decomposition. In the top row, points are colored by each image's mean pixel intensity; in the bottom row, points are colored by subject identity.

cluster membership, and a legend with the corresponding cluster summaries is added to the plot. We see that the embedding successfully captures the coupled low-dimensional structure, documents / authors within the same cluster, revealing cohesive groups. The visual separation between clusters indicates that the method effectively distinguishes between different thematic or collaborative areas, discovered from textual analysis (i.e., abstract or the paper or author names), while the embedding itself is derived from the tensor, which represents co-occurrence relationships.

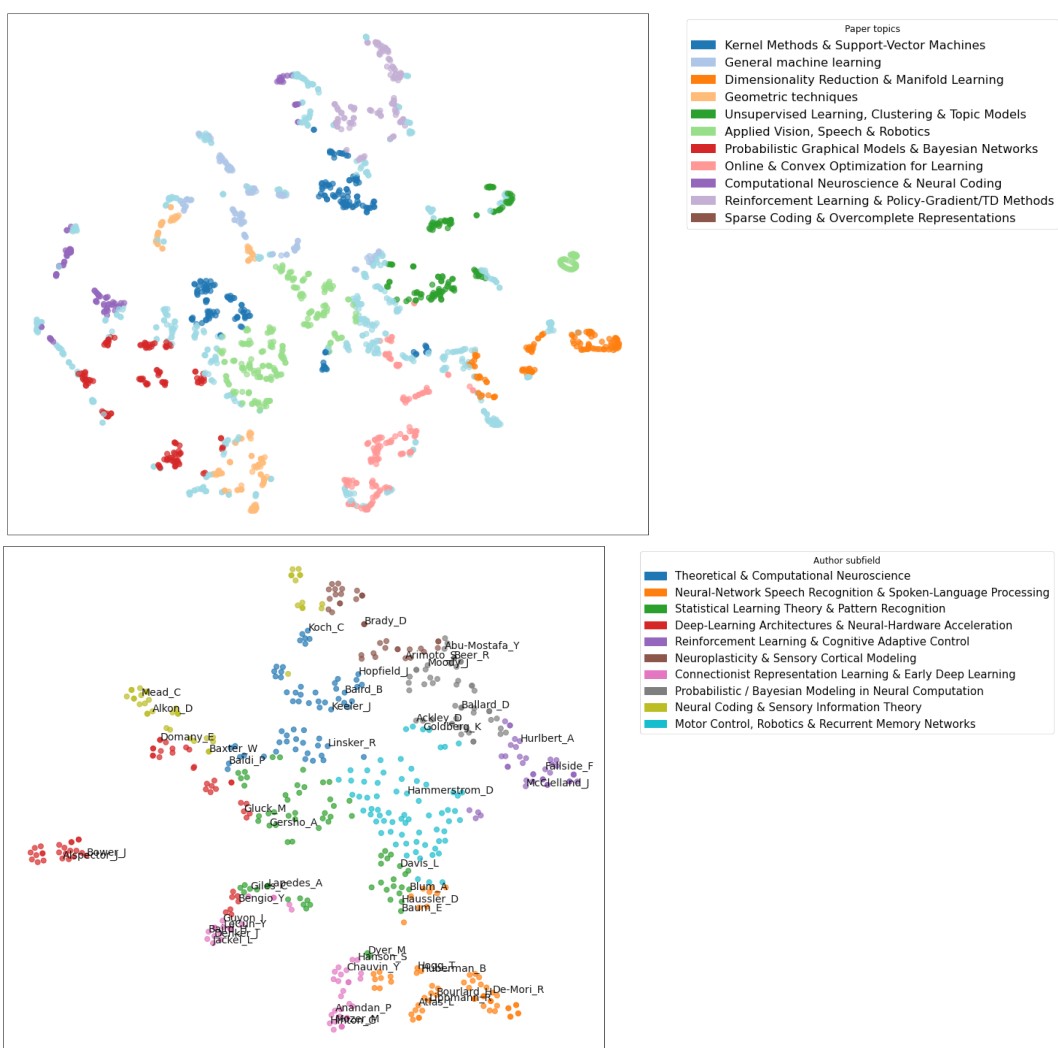

Figure 6: t-SNE embeddings of the NeurIPS publication tensor of words × papers × authors using the multiscale metric eq. (9). **Top:** Document embedding of documents. Colors indicate topic. **Bottom:** Author Embedding. Colors indicate specialization.

### A.3 Stability of compression

In Sec. 3.2 (full details in F), we describe a method to improve the computational complexity of TCML based on iterative compression (Yi et al., 2021). Here we verify the correctness of the compressed algorithm by evaluating its ARI score in comparison to TCML's on the time-varying linkage2 dataset (Fig. 7). We see that the compression strategy does not degrade the clustering accuracy too much, and in fact matches or outperforms vanilla TCML.

### A.4 Parameterization of tensor decomposition methods

In the figure below, Fig. 8, we illustrate the tradeoff for the tensor decomposition method between rank, reconstruction loss, and various qualitative measures on the Linkage 2 dataset under a low degree of noise (0.2).

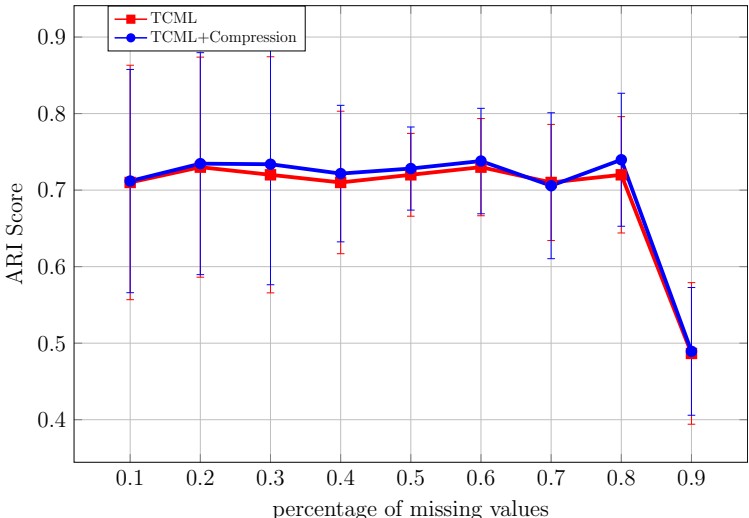

Figure 7: Evaluation of TCML + compression on a clustering task on the time-varying linkage2 dataset.

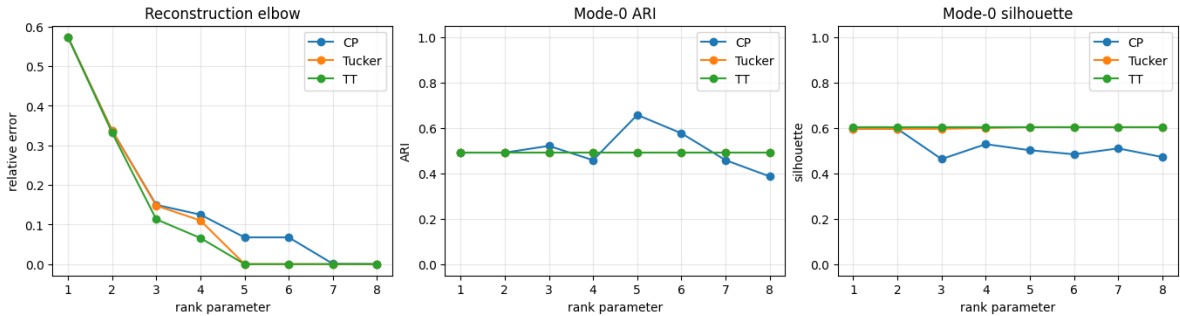

Figure 8: Rank selection based on reconstruction loss.

## B  Proof of Proposition 3.2

We derive the proximal mapping of $h(\mathcal{V}) = \upsilon\rho(\|\mathcal{V}\|_{\mathrm{F}})$ in two steps. First, recall that the $\mathcal{V}$-update can be decomposed into independent block updates on $\mathbf{v}_{d,l}$ by the same "one-dimensional shrink" $\mathrm{prox}_\rho$ on the group norm.

**Proposition B.1** (Groupwise proximal operator). *Let $\mathcal{G}$ be a partition on the index set $[G]$, namely for all $g \in G$, $\bigcup_{g \in G} = G$ and for all distinct $g, g' \in G$, $g$ and $g'$ are disjoint. Suppose the proximal mapping of a scalar function $\rho : \mathbb{R}_{\geq 0} \to \mathbb{R}$ is defined. Let $h(\mathbf{v}) = \sum_{g \in \mathcal{G}} \rho(\|\mathbf{v}_g\|_2)$, where each $\mathbf{v}_g \in \mathbb{R}^{|g|}$ is the sub-vector of $\mathbf{v}$ restricted to indices $g$. Then*

$$\big[\mathrm{prox}_h(\mathbf{v})\big]_g = \mathrm{prox}_\rho\big(\|\mathbf{v}_g\|_2\big) \, \frac{\mathbf{v}_g}{\|\mathbf{v}_g\|_2}.$$

The next proposition gives an analytical expression for the proximal mapping of the snowflake penalty.

**Proposition B.2** (Snowflake Proximal Map). *The proximal mapping of the snowflake penalty at $x$ is associated with a root $\phi^\star$ of the cubic function $\phi^3 + \epsilon\phi^2 - x\phi - x\epsilon + \frac{v}{2}$ that minimizes the original proximal objective over $z = \phi^2$.*

*Proof.* Note that the snowflake penalty can be expressed as

$$\rho(z) = \sqrt{z} - \epsilon \ln\big(\sqrt{z} + \epsilon\big) + \epsilon \ln \epsilon.$$

Then

$$
\begin{aligned}
\operatorname{prox}_{v\rho}(x) &= \underset{z \geq 0}{\arg\min}\ v\big[\sqrt{z} - \epsilon \ln(\sqrt{z} + \epsilon) + \epsilon \ln \epsilon\big] + \frac{1}{2}(z - x)^2 \\
&= \left[\underset{\phi \geq 0}{\arg\min}\ v[\phi - \epsilon \ln(\phi + \epsilon) + \epsilon \ln \epsilon] + \frac{1}{2}(\phi^2 - x)^2\right]^2.
\end{aligned}
$$

By the extreme value theorem, $\operatorname{prox}_{v\rho}(x)$ is either 0 or the square of a stationary point of

$$
f(\phi) = v[\phi - \epsilon \ln(\phi + \epsilon) + \epsilon \ln \epsilon] + \frac{1}{2}(\phi^2 - x)^2.
$$

The stationary points of $f(\phi)$ satisfy

$$
P(\phi) = \phi \left[\phi^3 + \epsilon\phi^2 - x\phi - \epsilon x + \frac{v}{2}\right] = 0.
$$

Thus,

$$
\operatorname{prox}_{v\rho}(x) = \left[\underset{\phi : P(\phi) = 0, \phi \geq 0}{\arg\min} \left\{v[\phi - \epsilon \ln(\phi + \epsilon) + \epsilon \ln \epsilon] + \frac{1}{2}(\phi^2 - x)^2\right\}\right]^2.
$$

Note that we only need to find the nonnegative roots of the cubic polynomial

$$
\phi^3 + \epsilon\phi^2 - x\phi - \epsilon x + \frac{v}{2}. \tag{11}
$$

We compute the roots using Cardano's formula and then take the square of the nonnegative roots to obtain candidate stationary points. Once we have the candidate stationary points, we evaluate $v\rho(z) + \frac{1}{2}(x - z)^2$ at 0 and the candidate stationary points and return the minimizer. □

Propositions B.1 and B.2 together prove Proposition 3.2.

## C Proof of Theorem 3.3

We study the nonconvex tensor-smoothing problem

$$
\min_{\mathcal{U} \in \mathbb{R}^{n_1 \times \cdots \times n_D}} F(\mathcal{U}) := \frac{1}{2}\|\mathcal{X} - \mathcal{U}\|_{\mathrm{F}}^2 + \sum_{d=1}^{D} \gamma_d \sum_{e \in E_d} \rho(\|\mathcal{U} \times_d \Delta_{d,e}\|_{\mathrm{F}}), \tag{12}
$$

where $\mathcal{X} \in \mathbb{R}^{n_1 \times \cdots \times n_D}$ and each graph $G_d$ is connected. Although $\rho$ is concave, the objective in eq. (12) admits a difference-of-convex decomposition.

Introduce $\mathcal{V}_{d,e} = \mathcal{U} \times_d \Delta_{d,e}$ and define the linear map $\mathcal{D} : \mathbb{R}^N \to \mathbb{R}^M$ ($M = \sum_d |E_d| n_{-d}$) by $\operatorname{vec}(\mathcal{V}) = \mathcal{D}\operatorname{vec}(\mathcal{U})$. The problem becomes

$$
\min_{\mathcal{U},\mathcal{V}} \frac{1}{2}\|\mathcal{X} - \mathcal{U}\|_{\mathrm{F}}^2 + \sum_{d,e} \gamma_d \rho(\|V_{d,e}\|_{\mathrm{F}}) \quad \text{s.t.} \quad \operatorname{vec}(V) - \mathcal{D}\operatorname{vec}(\mathcal{U}) = 0. \tag{13}
$$

We now state the main convergence results. The analysis follows the framework of Wang et al. (2019). We identify the Wang et al. (2019) blocks $(x_0, x_1, \ldots, x_p)$ with our tensor variables by setting

$$
x_0 \longleftrightarrow u = \operatorname{vec}(\mathcal{U}) \in \mathbb{R}^N, \quad x_i \longleftrightarrow v_{d,j} = \operatorname{vec}(\mathcal{V}_{d,e}) \in \mathbb{R}^{n-d},
$$

where $(d, j)$ runs over all edges in each mode. The quadratic term $\frac{1}{2}\|\mathcal{X} - \mathcal{U}\|_{\mathrm{F}}^2$ becomes $f_0(u) = \frac{1}{2}\|x - u\|_2^2$ with $x = \operatorname{vec}(\mathcal{X})$, and each penalty $\gamma_d \rho(\|\mathcal{V}_{d,e}\|_{\mathrm{F}})$ becomes $f_{d,j}(v_{d,j})$. The single linear constraint $\operatorname{vec}(V) - \mathcal{D}\operatorname{vec}(\mathcal{U}) = 0$ corresponds to the block-matrix system $[A_{0,d,j} \mid -I](u, v_{d,j}) = 0$.

In summary, our model corresponds with the slightly simpler model in their paper:

$$\underset{u,v_{1,1},\ldots,v_{D,|\mathcal{E}_D|}}{\text{minimize}} \quad \underbrace{f_0(u) + \sum_{d=1}^{D}\sum_{j=1}^{|\mathcal{E}_d|} f_{d,j}(v_{d,j})}_{\varphi(u,v)}$$

subject to

$$\underbrace{\begin{pmatrix} A_{0,1,1} & -I & 0 & 0 & \cdots & 0 \\ A_{0,1,2} & 0 & -I & 0 & \cdots & 0 \\ \vdots & \vdots & \vdots & \ddots & \vdots & \vdots \\ A_{0,D,|\mathcal{E}_D|-1} & 0 & \cdots & 0 & -I & 0 \\ A_{0,D,|\mathcal{E}_D|} & 0 & \cdots & 0 & 0 & -I \end{pmatrix}}_{A} \begin{pmatrix} u \\ v_{1,1} \\ v_{1,2} \\ \vdots \\ v_{D,|\mathcal{E}_D|-1} \\ v_{D,|\mathcal{E}_D|} \end{pmatrix} = 0.$$

Where the objective terms

$$f_0(u) = \frac{1}{2}\|x - u\|_2^2$$
$$f_{d,j}(v_{d,j}) = \gamma_d \rho(|v_{d,j}\|_2).$$

In this simpler optimization problem, it is necessary to verify assumptions A1, A3b, and A4 in Wang et al. (2019) to demonstrate the following convergence result:

**Theorem C.1** (Stationarity of ADMM limit points (Wang et al., 2019, Thm. 1))**.** *There exists a finite threshold $\bar{\beta} > 0$ such that for any penalty parameter $\beta > \bar{\beta}$ the ADMM iterates $\{(\mathcal{U}^k, \mathcal{V}^k, \lambda^k)\}_{k \geq 0}$ satisfy the following:*

1. ***Boundedness.*** *The sequence $\{(\mathcal{U}^k, \mathcal{V}^k, \lambda^k)\}$ is bounded.*

2. ***Stationary limit points.*** *Every cluster point $(\mathcal{U}^\star, \mathcal{V}^\star, \Lambda^\star)$ obeys*

$$\text{vec}(\mathcal{V}^\star) - \mathcal{D}\,\text{vec}(\mathcal{U}^\star) = 0, \qquad 0 \in \partial f_0\big(\text{vec}(\mathcal{U}^\star)\big) + \mathcal{D}^\top \lambda^\star, \qquad 0 \in \partial g\big(\text{vec}(\mathcal{V}^\star)\big) - \lambda^\star,$$

   *hence $(\mathcal{U}^\star, \mathcal{V}^\star)$ is a stationary (KKT) point of the equality-constrained problem eq. (13) and therefore of the original tensor-smoothing objective eq. (12).*

**Assumption C.2** (A1: Coercivity)**.** Let

$$\mathcal{F} = \big\{(u,v) \in \mathbb{R}^N \times \mathbb{R}^M : A(u,v) = 0\big\},$$

and define

$$\varphi(u,v) = f_0(u) + \sum_{d,j} f_{d,j}(v_{d,j}) = \tfrac{1}{2}\|x - u\|_2^2 + \sum_{d,j}\gamma_d\,\rho(\|v_{d,j}\|_2).$$

Then $\varphi$ is coercive on $\mathcal{F}$: whenever $(u^k, v^k) \in \mathcal{F}$ satisfies $\|(u^k, v^k)\| \to \infty$, one has $\varphi(u^k, v^k) \to \infty$.

*Proof.* Let $\{(u^k, v^k)\} \subset \mathcal{F}$ with $\|(u^k, v^k)\| \to \infty$. Then either $\|u^k\| \to \infty$ or $\|v^k\| \to \infty$. But $v^k = \mathcal{D}\,u^k$ implies $\|v^k\| \leq \|\mathcal{D}\|\,\|u^k\|$, so in fact $\|v^k\| \to \infty$ also forces $\|u^k\| \to \infty$. Hence in every case $\|u^k\| \to \infty$. Since

$$\varphi(u^k, v^k) \geq \tfrac{1}{2}\|x - u^k\|_2^2 = f_0(u^k),$$

and $f_0(u) = \tfrac{1}{2}\|x - u\|_2^2 \to \infty$ as $\|u\| \to \infty$, it follows that $\varphi(u^k, v^k) \to \infty$. $\qquad\square$

**Assumption C.3** (A3(b): Lipschitz sub-minimization paths)**.** Index the blocks as

$$x_0 = u, \quad x_i = v_{d,j}, \quad i = 1, \ldots, p,$$

and write $\varphi(u, v) = f_0(u) + \sum_{d,j} f_{d,j}(v_{d,j})$. There exists $M > 0$ such that for each $i = 0, \ldots, p$ the mapping

$$F_i : \delta \mapsto \arg\min_{x_i} \Big\{ \varphi(x_{<i}, x_i, x_{>i}) \; : \; A_i \, x_i = \delta \Big\}$$

is single-valued and $M$-Lipschitz on $\mathrm{Im}(A_i)$.

*Proof.*

$i = 0$ Here $A_0$ is the stack of all $A_{0,d,j}$ enforcing $A_0 \, u = \delta$. The subproblem

$$\min_u \; \tfrac{1}{2}\|x - u\|_2^2 \quad \text{s.t.} \quad A_0 \, u = \delta$$

is strictly convex and has a unique minimizer. Moreover one shows by the normal-equations that

$$F_0(\delta) = x \; + \; A_0^\top \big(A_0 A_0^\top\big)^\dagger (\delta - A_0 x),$$

which is an affine function of $\delta$. Hence $F_0$ is Lipschitz continuous on $\Im(A_0)$.

$i \geq 1$ For each difference-block $v_{d,j}$, $A_i = I$ so the constraint is $v_{d,j} = \delta$. The subproblem $\min_v \; \gamma_d \, \rho(\|v\|_2)$ s.t. $v = \delta$ clearly has the unique solution $v = \delta$, and the mapping $F_i(\delta) = \delta$ is 1-Lipschitz.

In summary, each block subproblem has a unique solution and the block-wise solution maps are Lipschitz, so A3(b) holds. $\square$

**Assumption C.4** (A4: Objective regularity)**.** Write

$$\varphi(u, v) \; = \; f_0(u) \; + \; \sum_{d,j} f_{d,j}(v_{d,j}),$$

with

$$f_0(u) = \tfrac{1}{2}\|x - u\|_2^2, \quad f_{d,j}(v) = \gamma_d \, \rho(\|v\|_2).$$

Then $f_0$ is lower-semicontinuous, and each $f_{d,j}$ is restricted prox-regular.

*Proof.*

1. $f_0(u) = \tfrac{1}{2}\|x - u\|_2^2$ is a quadratic, hence continuous (and $C^1$ with Lipschitz gradient), so lower-semicontinuous.

2. Fix any $(d, j)$. Since $\rho \in C^2[0, \infty)$ with $\rho''(z) = -\tfrac{1}{4}(z + \epsilon)^{-3/2} \geq -L_\rho$ for $L_\rho = 1/(4\epsilon^{3/2})$, standard calculus shows

$$\nabla^2 f_{d,j}(v) \; \succeq \; -\gamma_d L_\rho \, I \quad \text{for all } v \neq 0.$$

Hence

$$v \mapsto f_{d,j}(v) \; + \; \tfrac{\gamma_d L_\rho}{2}\|v\|_2^2$$

is convex on $\mathbb{R}^{n-d}$. By the characterization of prox-regularity (e.g. Rockafellar et al. (2009)) , this implies $f_{d,j}$ is prox-regular everywhere. Restricted prox-regularity follows immediately.

$\square$

## D  Additional Proofs

*Proof of Proposition B.1.* Consider the case when $\mathbf{x} \neq \mathbf{0}$.

$$\frac{1}{2}\|\mathbf{x} - \boldsymbol{\theta}\|_2^2 + \gamma\rho(\|\boldsymbol{\theta}\|_2) =$$

$$= \frac{1}{2}\|\mathbf{x}\|_2^2 - \langle \mathbf{x}, \boldsymbol{\theta} \rangle + \frac{1}{2}\|\boldsymbol{\theta}\|_2^2 + \gamma\rho(\|\boldsymbol{\theta}\|_2)$$

$$\geq \frac{1}{2}\|\mathbf{x}\|_2^2 - \|\mathbf{x}\|_2\|\boldsymbol{\theta}\|_2 + \frac{1}{2}\|\boldsymbol{\theta}\|_2^2 + \gamma\rho(\|\boldsymbol{\theta}\|_2)$$

$$= \frac{1}{2}\left(\|\mathbf{x}\|_2 - \|\boldsymbol{\theta}\|_2\right)^2 + \gamma\rho(\|\boldsymbol{\theta}\|_2).$$

The inequality becomes equality when $\boldsymbol{\theta} = \alpha \frac{\mathbf{x}}{\|\mathbf{x}\|_2}$ for $\alpha \geq 0$. Therefore,

$$\min_{\boldsymbol{\theta}} \frac{1}{2}\|\mathbf{x} - \boldsymbol{\theta}\|_2^2 + \gamma\rho(\|\boldsymbol{\theta}\|_2) = \min_{\alpha \geq 0} \frac{1}{2}\left(\|\mathbf{x}\|_2 - \alpha\right)^2 + \gamma\rho(\alpha).$$

Therefore, $\alpha = \text{prox}_{\gamma\rho}(\|\mathbf{x}\|_2)$ and consequently

$$\text{prox}_{\gamma\rho(\|\cdot\|_2)} = \begin{cases} \text{prox}_{\gamma\rho}(\|\mathbf{x}\|_2)\frac{\mathbf{x}}{\|\mathbf{x}\|_2} & \mathbf{x} \neq \mathbf{0} \\ \mathbf{0} & \text{otherwise.} \end{cases}$$

$\square$

## E  Co-Organizing an Incomplete Data Tensor

Suppose instead of observing the complete data tensor $\mathcal{X} \in \mathbb{R}^{n_1 \times \cdots \times n_D}$. we only observe a fraction of its entries. Let

$$\Theta \subseteq [n_1] \times [n_2] \times \cdots \times [n_D]$$

be the subset of indices for the observed entries, and denote by $P_\Theta$ the projection operator that zeroes out unobserved entries:

$$[P_\Theta(\mathcal{Y})]_{i_1\ldots i_D} = \begin{cases} y_{i_1\ldots i_D} & \text{if } (i_1, \ldots, i_D) \in \Theta \\ 0 & \text{otherwise.} \end{cases}$$

A natural co-organizing objective for the partially observed tensor is

$$\min_{\mathcal{U}} \frac{1}{2}\|P_\Theta(\mathcal{X}) - P_\Theta(\mathcal{U})\|_{\text{F}}^2 + \sum_{d=1}^{D} \gamma_d J_d(\mathcal{U}), \tag{14}$$

The augmented Lagrangian for the equality constrained version of eq. (14) is

$$\mathcal{L}(\mathcal{U}, \mathcal{V}, \lambda) = \frac{1}{2}\|P_\Theta(\mathcal{X}) - P_\Theta(\mathcal{U})\|_{\text{F}}^2$$

$$+ \sum_{d=1}^{D} \sum_{e \in \mathcal{E}_d} \left[ \gamma_d \rho(\|\mathcal{V}_{d,e}\|_F) + \langle \lambda_{d,e}, \mathcal{V}_{d,e} - \mathcal{U} \times_d \boldsymbol{\Delta}_{d,e} \rangle + \frac{\beta}{2}\|\mathcal{V}_{d,e} - \mathcal{U} \times_d \boldsymbol{\Delta}_{d,e}\|_{\text{F}}^2 \right]. \tag{15}$$

The $\mathcal{U}$-update is the solution to a slightly different linear system

$$\min_{\mathcal{U}} \frac{1}{2}\|P_\Theta(\mathcal{X}) - P_\Theta(\mathcal{U})\|_{\text{F}}^2 + \sum_{d=1}^{D} \sum_{e \in \mathcal{E}_d} \frac{\beta}{2}\left\|\mathcal{V}_{d,e}^{(k+1)} - \mathcal{U} \times_d \boldsymbol{\Delta}_{d,e} + \frac{1}{\beta}\lambda_{d,e}^{(k)}\right\|_{\text{F}}^2.$$

Vectorizing $\mathcal{U}$ as $\mathbf{u}$ as before, we obtain the linear system

$$\left[\mathbf{P}_\Theta + \beta \sum_{d=1}^D \mathbf{A}_d^\mathsf{T} \mathbf{A}_d\right] \mathbf{u}^{(k+1)} = \mathbf{P}_\Theta\,\mathbf{x} + \sum_{d=1}^D \mathbf{A}_d^\mathsf{T}\left[\beta \mathbf{v}_d^{(k+1)} + \boldsymbol{\lambda}_d^{(k)}\right], \tag{16}$$

where $\mathbf{P}_\Theta$ is a diagonal matrix whose diagonal entries are zero or one. The $i$th diagonal entry is one if the $i$th entry of $\text{vec}(\mathcal{X})$ is observed and is zero otherwise. So, in the case when the data is a 2-way array or matrix $\mathbf{X} \in \mathbb{R}^{n_1 \times n_2}$, $\mathbf{P}_\Theta \in \{0,1\}^{n\times n}$, where $n = n_1 n_2$, is a diagonal matrix with a 1 in the $k$th diagonal entry if the $k$th entry in the matrix (column major ordering) is observed and 0 otherwise. More explicitly, if $(i,j) \in \Theta$, then $\mathbf{P}_\Theta(k,k) = 1$ where $k = i + n_1(j-1)$. The following proposition ensures that the matrix

$$\mathbf{P}_\Theta + \beta \sum_{d=1}^D \mathbf{A}_d^\mathsf{T} \mathbf{A}_d$$

is invertible.

**Proposition E.1.** *Let $\mathbf{Z}_d$ for $d = 1, \ldots, D$ denote $D$ positive semidefinite matrices. Then $\text{Null}(\sum_{d=1}^D \mathbf{Z}_d) = \cap_{d=1}^D \text{Null}(Z_d)$.*

*Proof.* Suppose $\mathbf{v} \in \text{Null}(\sum_{d=1}^D \mathbf{Z}_d)$. Then

$$0 = \mathbf{v}^\mathsf{T}\left(\sum_{d=1}^D \mathbf{Z}_d\right)\mathbf{v} = \sum_{d=1}^D \mathbf{v}^\mathsf{T}\mathbf{Z}_d\mathbf{v}.$$

But since $\mathbf{Z}_d$ are all positive semidefinite, the quadratic forms $\mathbf{z}^\mathsf{T}\mathbf{Z}_d\mathbf{z} \geq 0$ for all $d$. A sum of nonnegative numbers is zero if and only if each of the summands is zero. Therefore, $\mathbf{v} \in \text{Null}(\mathbf{Z}_d)$ for all $d$.

Note that $\mathbf{u}^\mathsf{T}\mathbf{A}_d^\mathsf{T}\mathbf{A}_d\mathbf{u} = \text{tr}(\mathbf{U}_{(d)}^\mathsf{T}\boldsymbol{\Phi}_d^\mathsf{T}\boldsymbol{\Phi}_d\mathbf{U}_{(d)})$. Thus, $\mathbf{u}^\mathsf{T}\mathbf{A}_d^\mathsf{T}\mathbf{A}_d\mathbf{u} = 0$ if and only if the columns of $\mathbf{U}_{(d)}$ live in the span of the indicator function vectors of the connected components of the mode-$d$ graph encoded in $\boldsymbol{\Phi}_d$. This implies that $\nu\mathbf{P}_\Theta + \sum_{d=1}^D \mathbf{A}_d^\mathsf{T}\mathbf{A}_d$ is invertible if and only if

$$\mathbf{P}_\Theta(\boldsymbol{\chi}_{r_1}^{(1)} \circ \cdots \circ \boldsymbol{\chi}_{r_D}^{(D)}) \neq \mathbf{0}.$$

In other words, the matrix is invertible if and only if we observe at least one entry within the checkercubes defined by the connected components of the graphs associated with each mode. Note that this can be verified in linear time using a simple breadth first search on each mode. $\square$

### E.1 Multiscale metric and embedding

Suppose we have a partially observed data tensor $\mathcal{X} \in \mathbb{R}^{n_1 \times n_2 \times \cdots \times n_D}$, where some entries are missing. Let $\Theta \subseteq \prod_{d=1}^D \{1, \ldots, n_d\}$ be the index set of observed entries, and let $\Theta^c$ be the complement (unobserved entries).

**Smoothing at multiple scales.** For each mode $d$, we have a regularization parameter $\gamma_d$. Solving the *multiway* clustering or smoothing problem for $\{\gamma_d\}$ yields a smoothed tensor $\mathcal{U}^{(\ell_1,\ldots,\ell_D)}$, where $\ell_d$ indexes the grid point (or "scale") used for mode $d$. (Equivalently, $\ell_d$ could track a sequence of $\gamma_d$-values along mode $d$.)

Once we have $\mathcal{U}^{(\ell_1,\ldots,\ell_D)}$, we *impute* the missing entries of $\mathcal{X}$ by

$$\widetilde{\mathcal{X}}^{(\ell_1,\ldots,\ell_D)} = \mathcal{P}_\Theta(\mathcal{X}) + \mathcal{P}_{\Theta^c}(\mathcal{U}^{(\ell_1,\ldots,\ell_D)}),$$

where $\mathcal{P}_\Theta$ (resp. $\mathcal{P}_{\Theta^c}$) projects a tensor onto the observed (resp. unobserved) entries. Hence, $\widetilde{\mathcal{X}}^{(\ell_1,\ldots,\ell_D)}$ is a fully "filled-in" version of $\mathcal{X}$ at the chosen set of scales $\{\gamma_1, \ldots, \gamma_D\}$.

Note that if there are no missing values (i.e. if $\Theta^c = \emptyset$) then $\widetilde{\mathcal{X}}^{(\ell_1,\ldots,\ell_D)} = \mathcal{X}$ at every scale, so $\left\|\widetilde{\mathcal{X}}_{(i)}^{(\ell_1,\ldots,\ell_D)} - \widetilde{\mathcal{X}}_{(j)}^{(\ell_1,\ldots,\ell_D)}\right\|_2 = \left\|\mathcal{X}_{(i)} - \mathcal{X}_{(j)}\right\|_2$, Then eq. (6) and eq. (7) just apply a constant factor to the usual distance, summing across scale tuples. Likewise, if $\Theta^c \neq \emptyset$, each $\widetilde{\mathcal{X}}^{(\ell_1,\ldots,\ell_D)}$ uses a different "smooth guess" for unobserved values, guided by the multiway clustering/smoothing at that set of $\{\gamma_d\}$. Summing over many scales yields a robust measure that captures both local and global geometry.

## F    Problem compression

To reduce the cost of the ADMM subproblems, we adapt the *compression* strategy proposed in Yi et al. (2021), originally designed for vector-data, to tensor co-manifold learning. This strategy, novel in the context of tensor data, aggregates the original $D$-mode data tensor into a smaller compressed tensor, which leverages the clusters revealed at finer resolutions. Suppose each mode $d \in \{1, \ldots, D\}$ is partitioned into $k_d$ clusters, denoted $c_d[i] \subseteq [n_d]$ for $i \in [k_d]$. Let

$$\mathcal{X} \in \mathbb{R}^{n_1 \times n_2 \times \cdots \times n_D}$$

be the original data tensor, and let $\mathcal{W} \in \mathbb{R}_{\geq 0}^{n_1 \times \cdots \times n_D}$ denote a weight tensor. The *compressed* data tensor

$$\widetilde{\mathcal{X}} \in \mathbb{R}^{k_1 \times \cdots \times k_D}$$

is defined by a weighted average over each multi-dimensional block:

$$\widetilde{\mathcal{X}}_{i_1, \ldots, i_D} = \frac{\displaystyle\sum_{(m_1, \ldots, m_D) \in c_1[i_1] \times, \cdots, \times c_D[i_D]} \mathcal{W}_{m_1, \ldots, m_D}^2 \mathcal{X}_{m_1, \ldots, m_D}}{\displaystyle\sum_{(m_1, \ldots, m_D) \in c_1[i_1] \times, \cdots, \times c_D[i_D]} \mathcal{W}_{m_1, \ldots, m_D}^2}, \quad \text{for } i_d \in [k_d]. \tag{17}$$

Hence all entries in the same "checkercube" (the Cartesian product of mode-$d$ clusters) are aggregated into a single value $\widetilde{\mathcal{X}}_{i_1, \ldots, i_D}$. The weights $\mathcal{W}_{m_1, \ldots, m_D}$ emphasize or de-emphasize specific entries.

Once $\widetilde{\mathcal{X}}$ is formed, we solve a smaller *compressed* co-organizing problem:

$$\min_{\mathcal{U} \in \mathbb{R}^{k_1 \times \cdots \times k_D}} \frac{1}{2} \left\| \widehat{\mathcal{W}}^{1/2} \odot \left( \widetilde{\mathcal{X}} - \mathcal{U} \right) \right\|_{\mathrm{F}}^2 + \gamma \sum_{d=1}^{D} \sum_{e \in \widehat{\mathcal{E}}_d} \widehat{w}_{d,e}\, \rho \left( \left\| \mathcal{U} \times_d \widehat{\boldsymbol{\Delta}}_{d,e} \right\|_{\mathrm{F}} \right), \tag{18}$$

where $\mathcal{U} \in \mathbb{R}^{k_1 \times \cdots \times k_D}$ is the unknown in the compressed domain, $\widehat{\mathcal{W}} \in \mathbb{R}^{k_1 \times \cdots \times k_D}$ is the aggregated weight tensor, *i.e.*, $\widehat{\mathcal{W}}_{i_1, \ldots, i_D} = \sum_{(m_1, \ldots, m_D) \in c_1[i_1] \times \cdots \times c_D[i_D]} \mathcal{W}_{i_1, \ldots, i_D}^2$, and for each mode $d$, $\widehat{\mathcal{W}}_{d,e}$ are the aggregated edge weights linking clusters $(i, j) = e$ in the compressed mode-$d$ graph $\widehat{\mathcal{E}}_d$, and $\widehat{\boldsymbol{\Delta}}_{d,e}$ is the corresponding oriented difference operator (cf. Section 2).

By construction, each entry of $\mathcal{U}$ represents a blockwise-constant solution in the original tensor $\mathcal{X}$. Increasing $\gamma$ merges adjacent blocks in the cluster graphs, preserving the underlying co-clustering structure at different levels of resolution.

Compression eq. (17) can drastically reduce the size of eq. (18), since $k_d \ll n_d$ typically. Consequently, one can efficiently compute solutions across a range of $\gamma$ values using this compressed problem without loss of the cluster structure: once clusters in each mode are coarsened, the uniform block structure in $\mathcal{U}$ remains valid for higher $\gamma$ settings. This yields a notable speedup compared to operating on the full $n_1 \times \cdots \times n_D$ tensor without loss of clustering fidelity.

## G    Practical implementation considerations

All experiments were run on a Lambda machine with an Intel(R) Core(TM) i9-9960X CPU @ 3.10GHz with 125 Gb of memory.

**ADMM step size**    ADMM has one free parameter, the step size $\beta$, which influences the algorithm's rate of convergence. Our goal in this section is to automate this choice, by adaptively tuning a sequence, $\beta_k$, for optimal performance

We employ the residual balancing technique proposed in He et al. (2000); Boyd et al. (2011). Residual balancing is based on the following observation: increasing $\beta_k$ strengthens the penalty term, yielding smaller primal residuals but larger dual ones; conversely, decreasing $\beta_k$ leads to larger primal and smaller dual

residuals. As both residuals must be small at convergence, it makes sense to "balance" them, i.e., tune $\beta_k$ to keep both residuals of similar magnitude. A simple scheme for this goal is

$$\beta_{k+1} = \begin{cases} \eta\beta_k & \text{if } \|r_k\|_2 > \mu\|d_k\|_2 \\ \beta_k/\eta & \text{if } \|d_k\|_2 > \mu\|r_k\|_2 \\ \beta_k & \text{otherwise,} \end{cases} \tag{19}$$

with $\mu > 1$ and $\eta > 1$ Boyd et al. (2011). Note that ADMM with adaptive penalty is not guaranteed to converge, unless $\beta_k$ is fixed after a finite number of iterations He et al. (2000).

## H  Single-mode illustration & selection of $\gamma$ sequences

In this section, we introduce a strategy for selecting a sequence $\{(\gamma_1, \ldots, \gamma_D)\}$. First, we describe a lower bound on $\gamma$ in the single-mode case such that the smoothed representation is effectively constant—i.e., the representations for all data points are the same.

For some symmetric nonnegative matrix $w_{ij} \in \mathbb{R}^{n \times n}$ and fixed $\gamma \in \mathbb{R}_+$. For simplicity, consider the single-mode $\ell_2$ regularized convex clustering problem.

$$\min_{U \in \mathbb{R}^{n \times k}} \left\{ F_\gamma(U) := \frac{1}{2} \sum_{i=1}^n \|U_i - X_i\|_2^2 + \frac{\gamma}{2} \sum_{i,j=1}^n w_{ij}\|U_i - U_j\|_2 \right\} \tag{20}$$

The solution $U^*$ is continuous in $\gamma$, and each choice of $\gamma$ parameterizes one level of the cluster hierarchy. For $\gamma = 0$, the solution is exactly $U^* = X$. As $\gamma \to \infty$, the solutions $U^* \to \bar{x}_w$, some constant (e.g. the grand mean in the unweighted case).

Note that in the single-mode case, the intermediate centroids computed from ADMM (analogously, eq. (5)) $U_{ij}$ are minimizers of the following least squares problem

$$\frac{1}{2} \sum_{i=1}^n \|X_i - U_i\|_2^2 + \gamma \sum_{i,j=1}^n \frac{w_{ij}}{2}\| - U_i + U_j + V_{ij} + \frac{1}{\beta}\lambda_{ij}\|^2 \tag{21}$$

### H.1  Dual problem

Consider the following equivalent problem.

$$\boxed{\min_{U \in \mathbb{R}^{d \times n}} \frac{1}{2}\|U - X\|_F^2 + \gamma\|UD^\top\|_{2,1}} \tag{22}$$

Where $D \in \mathbb{R}^{m \times n}$ is the incidence matrix of the graph induced by the weights, i.e. $L = \operatorname{diag}(d) - W = D^\top D$ for degrees $d$ and graph Laplacian $L$.

*Remark* H.1. We refer to *quadratic programs in standard form* as problems that can be expressed in the following way:

$$\min_{x \in \mathbb{R}^n} x^\top A x - x^\top b \quad \text{s.t. } g(x) \leq \gamma \tag{23}$$

where $A \in \mathbb{R}^{m \times m}$ is symmetric, $b \in \mathbb{R}^m$, and $g(x) : \mathbb{R}^m \to \mathbb{R}$ is some quadratic function of $x$. The dual of eq. (20) can be expressed as a quadratic program in standard form. Introducing the constraint $V = UD^\top$ and corresponding dual variables $\Lambda \in \mathbb{R}^{d \times m}$. The Lagrangian is given by

$$L(U, \Lambda, V) = \frac{1}{2}\|U - X\|_F^2 + \gamma\|V\|_{2,1} + tr(\Lambda^\top(V - UD^\top)) \tag{24}$$

Minimizing analytically over $U$ and $V$ gives

$$U^* = X + \Lambda D \quad V^* = 0 \text{ if } \|\Lambda_t\|_2 \leq \gamma \tag{25}$$

Where the second term is derived from the convex conjugate of the $\ell_{2,1}$ norm. More concretely we have

$$
\sup_{\Lambda \in \mathbb{R}^{d \times m}} -\frac{1}{2}\|\Lambda D\|_{\mathrm{F}}^2 - tr(\Lambda^\top W D^\top)
$$
$$
+ \inf_{V \in \mathbb{R}^{d \times m}} \left\{\gamma |V|_{2,1} + tr(\Lambda^\top V)\right\}. \tag{26}
$$

in which the infimum problem becomes

$$
\inf_V \gamma\|V\|_{2,1} + tr(\Lambda^\top V)
$$
$$
\Leftrightarrow \inf_V \sum_{j=1}^m \gamma|v_j|_2 + u_j^\top v_j
$$
$$
\Leftrightarrow \inf_V \sum_{j=1}^m \gamma\left(\sup_{|z_j|_2 \leq 1} z_j^\top v_j\right) + \lambda_j^\top v_j
$$
$$
\Leftrightarrow \inf_V \sum_{j=1}^m \left(\inf_{|z_j|_2 \leq 1} (-\gamma z_j + \lambda_j)^\top v_j\right). \tag{27}
$$

where $v_j$ and $\lambda_j$ are the $j$th column of $V$ and $\Lambda$. By the definition of the dual norm conjugate, the infimum over $V$ evaluates to 0 if $\|\lambda_j\|_2 \leq \gamma$ for all j, and $-\infty$ otherwise. That is, $\forall j = 1, \ldots, m$, $\|\lambda_j\|_2 \leq \gamma$. This leads to the dual

$$
\min_{\Lambda \in \mathbb{R}^{d \times m}} G(\Lambda) = \frac{1}{2}\|\Lambda D\|_{\mathrm{F}}^2 + tr(\Lambda^\top X D^\top)
$$
$$
\text{s.t. } \|\lambda_t\|_2 \leq \gamma, \ t = 1, \ldots, m \tag{28}
$$

Let $A = DD^\top \in \mathbb{R}^{m \times m}$ and $B = DX^\top \in \mathbb{R}^{m \times d}$. We have the dual problem

$$
\min_{\Lambda \in \mathbb{R}^{d \times m}} G(\Lambda) = \frac{1}{2}tr(\Lambda A \Lambda^\top) + tr(\Lambda B)
$$
$$
\text{s.t. } \|\lambda_t\|_2 \leq \gamma, \ t = 1, \ldots, m \tag{29}
$$

where $\lambda_t$ is the $t$-th row of $\Lambda$.

**Efficient $U$ update.** Consider eq. (21). If $U \in \mathbb{R}^{n \times k}$ and $W \in \mathbb{R}^{n \times n}$ is the weight matrix for the edges $\{(i,j)\} \in \mathcal{E}$, then for each $i = 1, \ldots, n$ we gather the partial derivatives of the objective:

$$
X_i - U_i + \gamma \sum_{j=1}^n w_{ij}\left[U_j - U_i + V_{ij} + \tfrac{1}{\beta}\lambda_{ij}\right] = 0.
$$

Rearranging terms, one obtains a linear system of the form

$$
(\gamma D + I)U - \gamma W U = \gamma W \odot \left(V + \tfrac{1}{\beta}\lambda\right)1_n + X,
$$

where $D = \mathrm{diag}(d_1, \ldots, d_n)$ with $d_i = \sum_j w_{ij}$ and $\odot$ denotes elementwise multiplication if applicable. Define $\widetilde{D}_\gamma = \gamma D + I$ and $\widetilde{L}_\gamma = \widetilde{D}_\gamma - \gamma W$. Hence

$$
\widetilde{L}_\gamma U = \gamma W \odot \left(V + \tfrac{1}{\beta}\lambda\right)1_n + X.
$$

The matrix $\widetilde{L}_\gamma$ is *diagonally dominant* and often sparse ($W$ is sparse in many applications). Thus, one can solve this system in nearly-linear time using specialized Laplacian solvers (e.g.).

**Extension to multiple modes.** In the multi-mode tensor setting, each mode $d$ contributes a similar Laplacian term $\widetilde{L}^{(d)}$. Summing these leads to the system presented in the main text

$$\left(\mathbf{I} + \beta \sum_{d=1}^{D} \mathcal{L}^{(d)}\right), \text{vec}(\mathcal{U}) = \text{RHS}\left(\mathcal{X}, \{\mathcal{V}^{(k)}, \lambda^{(k)}\}\right),$$

with each $\mathcal{L}^{(d)}$ analogous to $\widetilde{L}_\gamma$ above. In practice, we solve this via *iterative* methods that exploit the block or Kronecker structure of $\sum_d \mathcal{L}^{(d)}$. The key advantage remains that each Laplacian $\mathcal{L}^{(d)}$ is diagonally dominant and sparse in real-world graphs, ensuring efficiency at scale.

## H.2 Skew symmetry of update

Let $d_i = \sum_j w_{ij}$. Consider the first order condition for the partial derivatives:

$$X_i - U_i + \gamma \sum_j w_{ij}(U_j - U_i + V_{ij} + \frac{1}{\beta}\lambda_{ij}) = 0 \tag{30}$$

Re-arranging, we get the system

$$U_i(\gamma \sum_j w_{ij} + 1) = \gamma \sum_j w_{ij}(U_j + V_{ij} + \frac{1}{\beta}\lambda_{ij}) + X_i \tag{31}$$

Define the diagonal matrix $\widetilde{D}_\gamma = \gamma D + I$, where $D = \text{diag}(d)$. Thus, we have the system

$$\widetilde{D}_\gamma U = \gamma W U + \gamma W \odot (V + \frac{1}{\beta}\lambda)1_n + X \tag{32}$$

Let $\widetilde{L}_\gamma = \widetilde{D}_\gamma - \gamma W$. Simplifying the above expression, we get

$$\widetilde{L}_\gamma U = \gamma W \odot (V + \frac{1}{\beta}\lambda)1_n + X. \tag{33}$$

## H.3 Lower bound on $\gamma_{\mathbf{max}}$

**Proposition H.2.** *We aim to find a lower bound for $\gamma$ such that the solution to the constrained dual problem (with $\gamma$) coincides with the solution to the unconstrained dual problem, i.e. the solution as $\gamma \to \infty$.*

*The unconstrained dual problem is:*

$$\min_\Lambda \left\{ G(\Lambda) := \frac{1}{2}tr(\Lambda A \Lambda^\top) + tr(\Lambda B) \right\} \tag{34}$$

*To find the minimizer $\Lambda^*$, solve for the first order condition: take the derivative of $G(\Lambda)$ with respect to $\Lambda$ and set it to zero:*

$$\Lambda^* = -B^\top A^\dagger = -X D^\top (DD^\top)^\dagger \tag{35}$$

*To ensure that the constrained dual solution coincides with the unconstrained one, the constraints $||\lambda_t^*||_2 \le \gamma$ must be inactive. This happens when:*

$$\gamma \ge ||\lambda_t^*||_2 \quad \forall t \tag{36}$$

*i.e.,*

$$\gamma \ge ||[XD^\top(DD^\top)^\dagger]_t||_2 = ||XD^\top[(DD^\top)^\dagger]_{:,t}||_2 \quad \forall t. \tag{37}$$

*One lower-bound is then given by:*

$$\gamma \ge \max_{t=1,\ldots,m} ||XD^\top[(DD^\top)^\dagger]_{:,t}||_2 \tag{38}$$

**Proposition H.3.** *Suppose $\gamma_d \geq \gamma_d^* = 2\sqrt{\epsilon}\max_{e \in \mathcal{E}_d} \|(X - \bar{X}) \times_d \Delta_{d,e}\|_F$ for $d \in [D]$. Then $0 \in \partial_C f(\bar{X})$, where $\partial_C f(\bar{X})$ denotes the Clarke subdifferential of $f$ and $\bar{X}$, i.e., the grand mean tensor $\bar{X}$ is a stationary point of the objective function when all $\gamma_d$ are sufficiently large.*

*Proof.* Note that the objective function is the sum of Clarke regular functions. Therefore, the Clarke subdifferential of the objective function is the sum of the Clarke subdifferentials of the summand functions that make up the objective, i.e.,

$$\partial_C F(U) \quad = \quad \{U - X\} + \sum_{d=1}^{D} \sum_{e \in \mathcal{E}_d} \gamma_d \, \partial_C \left(\rho(\|U \times_d \Delta_{d,e}\|_F)\right).$$

The Clarke chain rule gives

$$\partial_C \left(\rho(\|U \times_d \Delta_{d,e}\|_F)\right) \quad = \quad \begin{cases} \rho'(\|U \times_d \Delta_{d,e}\|_F)\frac{U \times_d \Delta_{d,e}}{\|U \times_d \Delta_{d,e}\|_F} & \text{if } U \times_d \Delta_{d,e} \neq 0 \\ \frac{1}{2\sqrt{\epsilon}}\{W : \|\,\|W \times_d \Delta_{d,e}\|_F \leq 1\} & \text{if } U \times_d \Delta_{d,e} = 0. \end{cases}$$

Therefore, the stationarity condition $0 \in \partial_C F(\bar{X})$ is equivalent to

$$X - \bar{X} \in \sum_{d=1}^{D} \sum_{e \in \mathcal{E}_d} \frac{\gamma_d}{2\sqrt{\epsilon}}\{W : \|W \times_d \Delta_{d,e}\|_F \leq 1\}.$$

The right-hand side is a convex set that scales linearly with $\gamma_d$, while $X - \bar{X}$ is a fixed tensor. We next show that $X - \bar{X}$ is contained in the convex set for sufficiently large $\gamma_d$.

Choose any nonnegative weights $\alpha_{d,e} > 0$ with $\sum_{d,e} \alpha_{d,e} = 1$, and set

$$W_{d,e} \quad = \quad \frac{2\sqrt{\epsilon}\,\alpha_{d,e}}{\gamma_d}\,(X - \bar{X}).$$

Note that

$$\sum_{d=1}^{D} \sum_{e \in \mathcal{E}_d} \frac{\gamma_d}{2\sqrt{\epsilon}}\,W_{d,e} \quad = \quad \sum_{d,e} \alpha_{d,e}\,(X - \bar{U}) \quad = \quad X - \bar{X}.$$

Consequently, $X - \bar{X}$ is contained in the convex set on the right-hand side if $\|W_{d,e} \times_d \Delta_{d,e}\|_F \leq 1$. If $\gamma_d \geq \gamma_d^* = 2\sqrt{\epsilon}\max_{e \in \mathcal{E}_d} \|(X - \bar{X}) \times_d \Delta_{d,e}\|_F$ for $d \in [D]$. Then,

$$\begin{aligned} \|W_{d,e} \times_d \Delta_{d,e}\|_F \quad &= \quad \frac{2\sqrt{\epsilon}\,\alpha_{d,e}}{\gamma_d} \cdot \|(X - \bar{X}) \times_d \Delta_{d,e}\|_F \\ &\leq \quad \frac{\alpha_{d,e}\|(X - \bar{X}) \times_d \Delta_{d,e}\|_F}{\max_{e \in \mathcal{E}_d} \|(X - \bar{X}) \times_d \Delta_{d,e}\|_F} \\ &\leq \quad 1. \end{aligned}$$

## H.4   Selection of $\gamma$ sequences

Building on the analysis of determining $\gamma_{\max}$ for a single mode, we use this in order to determine the sequence of $\gamma$s in the multiway case.

We propose a strategy to select a sequence of $\{(\gamma_1, \ldots, \gamma_D)\}$ to smooth the tensor at multiple resolutions. We define, for each mode $d$, a value $\gamma_{d,max}$ beyond which the solution collapses that entire mode into a single cluster (see Proposition H.2 for a mode-specific bound). We then create a grid of $\gamma_d$ values, spanning $[0, \gamma_{d,\max}]$. Beginning with small initial values, we apply our tensor-smoothing algorithm (Algorithm 1) to obtain

a smooth estimate $\mathcal{U}(\gamma_1, \ldots, \gamma_D)$. Next, we fix all but one mode's parameter and increase the current mode's $\gamma_d$ in powers of two, each time refitting, until that mode merges to a single cluster. We then advance to the next mode and repeat, systematically covering a range of smoothing scales across all modes. Ultimately, when all $\gamma_d$ reach $\gamma_{d,\max}$, the tensor collapses into a single global cluster. Throughout this procedure, we store each estimated $\mathcal{U}(\gamma_1, \ldots, \gamma_D)$, yielding a collection of solutions for different co-clustering granularities along every mode.

