# OpenReview forum: "Multiscale Co-Manifold Learning on Tensors"
_TMLR — Rejected by TMLR_

### Review · Reviewer_gaKa · 2026-04-23

**Summary Of Contributions:**

The paper proposes Tensor Co-Manifold Learning (TCML), generalizing the matrix co-manifold learning of Mishne et al. (2019) to higher-order tensors. Contributions: (i) a family of convex-relaxed co-clustering problems with a concave (snowflake) group penalty on mode-wise graph edges, parameterized by per-mode weights $\gamma_d$; (ii) a multiscale slice-wise dissimilarity coupling each mode's geometry through smoothing on the others and (iii) an ADMM solver with claimed nearly linear per iteration cost and a convergence theorem (Thm. 3.3) via Wang Yin Zeng (2019)

**Strengths**. Clean, natural extension of Mishne et al. (2019). The multiscale distance construction is conceptually elegant. The sparse graph ADMM plus compression trick has independent algorithmic value and is the most reusable piece of the paper.

**Weaknesses**. Theorem 3.3 is stated for fixed $\beta$, while App. G admits the experiments use adaptive $\beta$ which lacks a guarantee. The nearly linear per iteration claim relies on an unspecified PCG preconditioner with no bound on CG iterations. Prop. H.2's collapse threshold is derived for $\ell_2$ but applied to schedule $\gamma$ for the snowflake penalized problem. Several hyperparameters ($\alpha$ in Eq. 6, the $\gamma$ grid, kNN graph $k$) are under specified and no code is released. Fig. 3 has one curve per method, no error bars, no seed count. Table 2's scaling trend is internally inconsistent. Fig. 6's evaluation is partly circular.

**Additional Comments:**

No additional comments.

**Audience:**

Yes

**Audience Explanation:**

Nonlinear manifold learning on tensors that preserves cross-mode coupling is relevant to a real slice of the TMLR audience. The sparse-graph ADMM for convex co-clustering of tensors, together with the compression-based acceleration, is of independent algorithmic value and likely to be reused. The related-work positioning (Mishne 2019, Chi 2020, Wu 2016, Stanley 2020) is accurate. With hyperparameters specified, code released, and the convergence and complexity claims tightened, this would be a worthwhile TMLR contribution.

**Broader Impact Concerns:**

No concerns. TCML is an unsupervised geometric/manifold-learning method operating on generic numeric tensors; there is no evident dual-use, fairness, or deployment concern that would warrant a Broader Impact Statement beyond TMLR's standard policy.

**Claims And Evidence:**

No

**Claims Explanation:**

**1. Thm. 3.3 does not cover the algorithm actually run.** Assumptions A2 and A5 of Wang–Yin–Zeng (2019) are not verified in App. C (they hold trivially since $A_p = -I$, but should be stated). More substantively, Thm. 3.3 requires fixed $\beta \ge \bar\beta$, while experiments use residual-balancing adaptive $\beta$. App. G itself concedes adaptive $\beta$ lacks a convergence guarantee unless frozen after finitely many iterations. The paper never says whether $\beta$ is ever frozen, so Thm. 3.3- strictly interpreted - does not apply to the algorithm that produced the reported results.

**2. The "nearly linear per-iteration" claim is unsubstantiated.** The V- and $\lambda$-updates are genuinely $O(\hat{M})$, but the U-update solves an SPD system via PCG at cost $k_{\text{CG}} \cdot O(N + \hat{M})$, and $k_{\text{CG}}$ is never bounded. The Spielman–Teng citation is aspirational; the paper says plain PCG is used, with no preconditioner described. Without a bound on $k_{\text{CG}}$ as $N$ grows, the phrasing overstates what is shown.

**3. Prop. H.2 is derived for the $\ell_2$-penalized problem but applied to schedule $\gamma$ for the snowflake (folded-concave) problem.** For folded-concave penalties the collapse threshold generally differs; the paper transfers the bound silently.

**4. Empirical claims are thinly supported.** Fig. 3 shows one curve per method, no error bars, no stated seed count, and no specification of $\alpha$ (Eq. 6), the $\gamma$ grid, the imputation scheme, or the kNN graph. For a method whose central claim is robustness under missing data, this is insufficient to back the Section 1 framing that TCML "outperforms existing methods".

**5. Table 2's scaling trend is internally inconsistent.** ADMM scales linearly in $n$, MM sublinearly (~$5.7\times$ per $10\times$), so the ADMM/MM speedup *shrinks* with $n$: ~$38\times$ at $n = 10^3$ vs. ~$4\times$ at $n = 10^6$. MM with a per-iteration linear solve should not scale sublinearly in $n$; without more setup detail, the asymptotic trend is not reliable.

**6. Real-data evaluation is qualitative and partly circular.** Table 3 has no error bars, no repeats, no code. Fig. 6 annotates the NeurIPS tensor embedding with cluster labels generated by an LLM from author names and paper titles - i.e., precisely the thematic information the method is supposed to recover from the co-occurrence tensor. The figure shows that when an embedding is colored by labels computed separately from adjacent side information, it appears clustered - which is not evidence that the embedding recovered the clusters.

All six items are revisable, but collectively they mean the paper's central claims (convergence, complexity, empirical superiority) are stated more strongly than the evidence supports.

**Requested Changes:**

**Critical to acceptance**

1. **Clarify Theprem. 3.3's applicability.** State explicitly whether $\beta$ is ever frozen during the reported runs. If not, soften Thm. 3.3 to an empirical convergence statement, or supply residual traces at several fixed $\beta$ values. Add a one-line verification of A2 and A5 of Wang–Yin–Zeng (trivial since $A_p = -I$, but worth stating).

2. **Substantiate or soften the "nearly linear per-iteration" claim.** Either describe the PCG preconditioner and show $k_{\text{CG}}$ remains bounded as $N$ grows on the Table 2 setup, or restrict the claim to a single matrix-vector multiply and report CG iteration counts separately.

3. **Address the $\ell_2$-vs-snowflake gap in Prop. H.2.** Derive a snowflake-specific collapse threshold, or state explicitly in the main text that Prop. H.2 is used as a heuristic proxy.

4. **Release a reference implementation and specify hyperparameters.** At minimum: the tensor ADMM solver, multiscale-distance aggregator, $\gamma$-schedule, and adaptive-$\beta$ rule. Per experiment, state the $\gamma$ grid, $\alpha$ in Eq. (6), the graph construction (type, $k$, weights), ADMM $(\beta, \mu, \eta, \text{max iter}, \text{tol})$, and imputation procedure.

---

> ### Author Response · Authors · 2026-06-07
> **Part 1 of response**
>
> We thank the reviewer for the careful and technically precise assessment of our work. We appreciate the positive comments on the multiscale distance construction and on the sparse-graph ADMM/compression components. We agree that several claims in the submitted version should be stated more carefully and that some experimental details were under-specified. Below we respond point-by-point and describe the revisions we have made.
>
> ### 1. Applicability of Theorem 3.3 and adaptive $\beta$
>
> We agree that Theorem 3.3, as stated, applies to the fixed-$\beta$ version of Algorithm 1. The residual-balancing $\beta$ rule used in Appendix G is a practical acceleration heuristic following He et al. (2000) and Boyd et al. (2011), whereas the convergence proof follows the fixed-penalty framework of Wang--Yin--Zeng (2019).
>
> In our splitting scheme, each auxiliary block is subject to a constraint of the form
> $$
> V_{d,e} - U \times_d \Delta_{d,e} = 0,
> $$
> and the block multiplying $V_{d,e}$ is $-I$. Hence the relevant full-rank assumptions are immediate.
>
> ### 2. “Nearly linear per-iteration” complexity claim
>
> We agree that the original claim may be too strong. We clarify that the $V$- and $\lambda$-updates are $O(M_c)$, and that one matrix-vector multiplication with the $U$-update operator costs $O(N+M_c)$, where
> $$
> N=\prod_d n_d, \qquad M_c=\sum_d |E_d|n_{-d}.
> $$
> For sparse mode graphs, $M_c=O(DN)$. The full PCG solve costs $O(q_k(N+M_c))$, where $q_k$ is the number of PCG iterations at ADMM iteration $k$. Thus the overall $U$-update is nearly linear only when $q_k$ is bounded, or when a nearly-linear SDD preconditioned method is applied.
>
> We have revised the main text to reflect this (see Section 3.1).
>
> ### 3. Use of Proposition H.2 for the snowflake penalty
>
> We agree that Proposition H.2, as written, is derived for the convex $\ell_{2,1}$-penalized single-mode problem. We have revised Appendix H and the main text accordingly.
>
> In particular, we have included a revised proof that is more direct. Specifically, the new result shows that the grand mean tensor of the data, i.e., the tensor whose elements are all the average value of the data tensor, is a stationary point of the snowflake-penalized objective function. The proof relies on Clarke subdifferential calculus since the snowflake penalty renders the optimization problem in the smoothing step non-convex. The objective function, however, is the sum of Clarke regular functions, which makes things relatively straightforward to show that 0 is in the Clarke subdifferential of the objective function. Our new result also provides an explicit lower bound on \gamma_d for all d \in [D] at which all subarrays along mode d of the solution tensor are identical.
>
> This clarification does not affect the empirical method: TCML only requires a grid spanning multiple smoothing regimes, not an exact analytic $\gamma_{\max}$. In the implementation, the schedule is refined by increasing $\gamma_d$ until the corresponding mode empirically merges. We have made this explicit in the revised manuscript.
>
> ### 4. Hyperparameters, missing-data experiments, error bars, and code
>
> We thank the reviewer for pointing out this oversight and have now added a reference implementation to the supplement and will add hyperparameter details.
> In the revised manuscript we have also add standard deviation to Tables 1 and 3, and thank the reviewer for pointing out our omission here.
>
> ### 5. Table 2 scaling trend
>
> We agree that Table 2 should not be used to infer asymptotic complexity exponents. The table reports end-to-end wall-clock times for complete solver runs, including warm starts, stopping behavior, implementation overhead, and, in the compressed variant, changing problem sizes along the path. It was intended as a practical runtime comparison, not as a log-log empirical proof of asymptotic scaling.
>
> The revised claim is narrower and more accurate: the proposed ADMM exploits sparse mode graphs to make the $V$- and $\lambda$-updates linear in the edge-indexed variables and each $U$-update matrix-vector multiplication nearly linear in the tensor size; the observed wall-clock speedup over the MM baseline is empirical and depends on solver tolerances, warm starts, and the inner linear-system iterations.

---

> ### Author Response · Authors · 2026-06-07
> **Part 2**
>
> ### 6. Real-data evaluation and Fig. 6
>
> We agree that the NeurIPS experiment in Fig. 6 should be described as qualitative. We wish to clarify the LLM-generated labels were not used to compute the TCML embedding; they were added only after the embedding from side information that was not used in calculating the embedding. The labels were intended to summarize visible regions of the plot, and thus provide qualitative evidence that the embedding recovers a meaningful organization of the data. However, the reviewer is correct that coloring an embedding by labels derived from author names and paper titles is not, by itself, a quantitative evaluation of clustering.
>
> We have revised the claim around Fig. 6 accordingly.
>
> For YaleFaces, we have added repetitions over independent pixel masks and now report mean $\pm$ standard error for trustworthiness, angle-intensity correlation, and individual silhouette.

---

### Review · Reviewer_y4M4 · 2026-05-20

**Summary Of Contributions:**

The authors propose TCML, a framework for simultaneous nonlinear dimensionality reduction across all modes of a tensor. The core idea is to smooth a data tensor at multiple scales via graph-regularized optimization, derive a multiscale distance between tensor slices that accounts for inter-mode coupling. The authors claimed that the main technical contribution is an ADMM algorithm for the tensor smoothing, replacing the Majorization-Minimization (MM) approach of Mishne et al. (2019) which scales poorly to higher-order tensors.

**Additional Comments:**

The paper requires substantial revision in its positioning, literature review, baseline selection, and writing before it can be considered for publication.

**Audience:**

No

**Audience Explanation:**

Low-rank tensor factorization for dimensionality reduction is well developed, and both nonlinear extensions and multiscale tensor decompositions exist in the literature. Without a thorough discussion of how TCML relates to and improves upon these lines of work, I expect that TMLR's audience would not be interested in this work.

**Claims And Evidence:**

No

**Claims Explanation:**

There are many tensor-based dimensionality reduction methods that account for coupling among tensor modes, e.g, tensor low-rank decomposition. The factor matrices (components) from such decompositions can serve as embeddings in a natural way. Without a clear explanation of why these approaches are insufficient, the necessity of the proposed framework remains unclear. I agree that the majority of these existing tensor-based dimensionality reduction methods are linear modeling, but as a shared common sense in the tensor community, tensors can easily have a really high-dimensional structure, and such high-dimensionality makes the tensor model well fit real datasets even with non-linear properties. The authors failed to explain why they need a non-linear setting. If the non-linearity of the model is essential, then the baseline needs to include non-linear tensor-based methods, e.g., Kernel CP, tensor-based Gaussian process, and tensor factorization neural network (See Section 10 in [1]).

[1] Liu, Yipeng, et al. Tensor computation for data analysis. Berlin: Springer, 2022.

In their experiments, they demonstrated that their methods are superior to linear low-rank approximation. However, they do not describe the rank-tuning at all, which does not make the experimental results convincing.


In addition, several overly strong claims lack supporting evidence:

> Reliable tools from matrix analysis typically fall short. (in Section 1)

This requires elaboration, i.e., which tools, why they fail, and how the proposed method overcomes these difficulties.

> Traditional approaches to dimension reduction and manifold learning for tensors focus on a single mode of the tensor.

This is not accurate. Tensor network methods such as Tensor Train and DMRG explicitly capture inter-mode dependencies through shared bond dimensions. While the authors note that such methods are linear, real data often admits (multi-)linear low-rank structure that (multi-)linear models can approximate well.


> Mishne et al. (2019) used a MM algorithm. This strategy, however, scales prohibitively poorly to higher-order tensors.

Since many tensor methods based on MM, this claim needs a more careful discussion, ideally with references to the general computational complexity of ADMM versus MM, to justify why ADMM is essential here.

**Requested Changes:**

1. I request a major update in the introduction so that readers can understand the motivation and contribution easily. In the introduction, the second and third paragraphs are largely redundant, both just arguing that mode-wise methods miss inter-mode coupling. The subsequent paragraph describes too many concepts (smoothing, multiscale metrics, kernel embeddings), and I failed to grasp the essence or key of the proposed method. I suggest including a conceptual figure to describe the authors' method.

2. Include nonlinear tensor decomposition methods as baselines and report the rank selection procedure.

3. Please include the description of why they avoid the simple batch-based scalable gradient-based method and developed ADMM (I assume reducing the hyper-parameter tuning for learning rate)

4. Please improve the paper structure. TCML as a framework is introduced in Section 4, yet Section 3.2 refers to "In this section, we present extension of TCML", which is confusing. A brief roadmap at the end of the introduction would help.

5. If the choice of penalty function satisfying Assumption 3.1 is not unique, explain why eq. 4 is preferred and how ε is selected in practice (it does not appear as an input to Algorithm 1).

6. I am not an expert in ADMM, but I assume ADMM is the method for convex optimization problems. Is eq(2) a convex optimization problem? Why the concavity of \rho makes the optimization difficult.

Minor points:
- Equation referencing style is inconsistent: sometimes "(1)", sometimes "eq. (1)". Please unify.
- Text in figures is too small to read comfortably.
- Several terms are used without adequate definition (e.g., "co-manifold," "degrees" in the first paragraph of Section 3).
- The color coding in Figure 1 is not explained. Does it represent tensor values?

---

> ### Author Response · Authors · 2026-06-07
> **Part 1 of response**
>
> We thank the reviewer for the careful assessment. Below we respond to each of your comments and describe the revisions we have made.
>
> We respectfully disagree that existing tensor decomposition methods make TCML unnecessary. Tensor decompositions and tensor networks are important and powerful, but they solve a different primary problem: learning multilinear or tensor-network factors, often for reconstruction or prediction. Our approach learns coupled, multiscale distances for each mode and uses those distances to recover nonlinear geometric structure. The experimental results show cases where this distinction is important, and using TCML outperforms tensor decomposition.
>
> We also emphasize that the sparse-graph ADMM and multiway compression components are not merely implementation details for the multiscale distance calculation but rather they provide a reusable solver for graph-regularized tensor co-clustering/smoothing problems, which is an independent contribution of the paper, which we have now highlighted in the introduction.
>
> Thus we believe our paper is of interest to the TMLR community.
>
>
> ### 1. Expanded discussion on tensor decomposition methods
>
> We thank the reviewer for raising this point. We agree that tensor decompositions, tensor networks, and related multilinear methods explicitly couple tensor modes, and we have revised the introduction and related-work section to state this. We also highlight that TCML is not designed primarily as a reconstruction or factorization method. It is designed to construct **mode-wise distances/geometries** whose local and global structure is informed by all other tensor modes.
>
> This difference is important in settings where the object of interest is not only a low-rank approximation of the tensor, but the intrinsic geometry of each mode: for example, whether time is recovered as a circle, whether samples on a helix are recovered as a one-dimensional curve, whether clustered samples remain separated under missingness, or whether face images organize simultaneously by illumination and identity. A CP/Tucker/TT factor matrix can certainly be used as an embedding, but the factors are optimized for a multilinear reconstruction objective. They are not explicitly optimized to produce coupled slice-wise distances that preserve nonlinear manifold structure.
>
> We have therefore revised the paper to state the claim more precisely and more strongly:
>
> Experimentally, we demonstrate that this distinction is important. In the time-varying linkage experiments, the tensor modes contain nonlinear and periodic latent structures: a helix, a curved surface or clustered mode, and a circular time variable. TCML recovers the coupled cluster/helix/time structure, including the circular time embedding, whereas CP and single-mode Diffusion Maps fail on at least one of these mode geometries. In the missing-data experiment, TCML gives the best clustering result across the missingness range and degrades only at extreme missingness, while CP degrades earlier. In YaleFaces, TCML achieves the strongest reported individual-separation score under both 40% and 60% pixel masking while also preserving the illumination organization. These results show that the proposed nonlinear, distance-based geometry is useful even though multilinear tensor methods are strong and appropriate baselines.

---

> ### Author Response · Authors · 2026-06-07
> **Part 2**
>
> ### 2. Nonlinear geometry
>
> We agree that the introduction tried to introduce too many concepts at once. We have rewritten the motivation to state the core idea more directly:
>
> This is the key concept that motivates our method. The smoothing step supplies a family of denoised/coarsened tensors. The distance step asks whether two mode-$d$ slices remain close across these smoothed versions. The embedding step is then standard: once a mode-wise distance matrix has been constructed, one may feed it to Diffusion Maps, t-SNE, UMAP, or another graph/kernel embedding method.
>
> Nonlinear geometry is needed because, as in the matrix case, the latent structures we target are often not well represented by a single linear subspace or multilinear factors. In our synthetic examples, the relevant mode geometries include a helix, a curved surface, clusters, and periodic time. TCML recovers the latent parametrization of these as nonlinear geometries: a curve for the helix, a curved/grid-like embedding for the surface, separated clusters for the clustered mode, and a circle for time. In contrast, CP and single-mode Diffusion Maps miss some of these geometries, especially the periodic time mode and in the missing-data regime.
>
> The same point is evident in real data. YaleFaces has two interpretable geometric factors: illumination varies continuously, while identity induces a class structure. TCML preserves the illumination organization while improving subject separation under severe pixel masking. Thus, the nonlinear embedding is not an aesthetic visualization choice; it is the final step after constructing a coupled distance that is intended to preserve curved, periodic, clustered, and multiscale structure.
>
> We have also clarified the phrase “distance/geometric structure”: by this we mean that the target object is a meaningful pairwise dissimilarity or neighborhood graph for each mode, from which one can recover continuous latent coordinates, clusters, or visualizations.
>
> ### 3. Baselines and rank selection
>
> We agree that the rank-selection protocol for tensor-decomposition baselines should be reported. We clarify that each tensor method was tuned via the elbow method.
>
> ### 4. Clarifying claims in the introduction
>
> We agree that several statements in the introduction are too broad. We have revised these claims.
>
> The intended point was that tensor problems often cannot be reduced to matrix problems without losing multiway structure, and that tensor rank, decomposition, identifiability, and computation differ substantially from the matrix case.
>
> We also clarify the statement that the MM strategy of Mishne et al. (2019) “scales prohibitively poorly to higher-order tensors.” More concretely,
>
> “A direct MM implementation for this graph-regularized co-clustering objective becomes expensive for the tensor-scale problems considered here, because each update requires operations over all edge-indexed mode differences. Our ADMM formulation exposes separable edge-wise proximal updates and a structured Kronecker-sum linear system, which can be applied implicitly using sparse mode graphs.”

---

> ### Author Response · Authors · 2026-06-07
> **Part 3**
>
> ### 6. Convexity, concavity of $\rho$, and applicability of ADMM
>
> We appreciate the reviewer’s question. Equation (2) is generally not convex under the snowflake penalty because $\rho$ is concave. More precisely, the fidelity term is convex, but the composition
> $$
> U \mapsto \rho(\|U\times_d \Delta_{d,e}\|_F)
> $$
> is generally nonconvex when $\rho$ is concave. The concavity of $\rho$ results in an objective which penalizes small differences strongly while penalizing large differences less severely. This reduces over-shrinkage of genuinely distinct tensor slices, analogously to folded-concave penalties such as MCP or SCAD.
>
> We have clarified that ADMM is not restricted to convex optimization in this paper. Note that our convergence result is based on the nonconvex nonsmooth ADMM framework of Wang, Yin, and Zeng (2019), and it establishes convergence to stationary/KKT points under the stated assumptions and fixed sufficiently large $\beta$. We have revised Section 3.1 to state this clearly.
>
> ### 7. Choice of snowflake penalty and selection of $\epsilon$
>
> We agree that the choice of penalty and $\epsilon$ should be explained. We use the snowflake penalty because it has the three following useful properties for the proposed ADMM solver:
>
> 1. it satisfies Assumption 3.1.
> 2. it behaves like a folded-concave group-fusion penalty, strongly encouraging small pairwise differences to merge while reducing shrinkage of large differences.
> 3. its proximal update has a closed-form reduction to a one-dimensional algebraic problem, which makes the edge-wise $V$-updates efficient.
>
> Other folded-concave penalties, such as MCP, SCAD, or logarithmic penalties, could also be used. The snowflake penalty is chosen for computational convenience. For all experiments, we set epsilon to
> $$
> 	\epsilon = 1e-3
> $$
>
> We have also added a snowflake-specific derivation of a terminal $\gamma$-scale in Appendix H.3. This result shows that, once the mode-wise $\gamma_d$ certain thresholds, the collapsed grand-mean tensor is a Clarke-stationary point of the snowflake-penalized objective. This gives a principled upper endpoint for the multiscale $\gamma$-grid and also clarifies the role of $\epsilon$, since the endpoint depends on $1/\rho'(0+)$.
>
> ### 8. Paper structure
>
> We apologize for the confusion regarding Section 3.2, the extensions we introduce are of the multiscale tensor smoothing approach introduced in section 3.1. Referring to this as extensions of TCML was a typo in that sense. We now write  “In this section, we present two extensions of our multiscale tensor smoothing algorithm to the large-scale and missing data regimes.”
> We have also added a roadmap paragraph at the end of the introduction.
>
> ### 9. Overview figure
>
> We thank the reviewer for this suggestion and have now replaced figure 1 with a  more comprehensive figure illustrating tensor smoothing at multiple scales (Fig.1 a) and the the TCML approach for one mode of the tensor, demonstrating how a single slice smoothed at multiple scales (Fig.1 b)  is used in pairwise slice distance (Fig.1 c) which are then used to recover the underlying nonlinear geometry  (Fig.1 d).

---

> > ### Comment · Reviewer_y4M4 · 2026-06-07
> >
> > I appreciate that the authors have toned down the overclaiming statements, replaced Figure 1, and clarified the convexity/ADMM discussion. However, I still do not find the response fully convincing for some my concerns, and I will maintain my current evaluation.
> >
> > --
> >
> > >Tensor decompositions and tensor networks are important and powerful, but they solve a different primary problem: learning multilinear or tensor-network factors, often for reconstruction or prediction. Our approach learns coupled, multiscale distances for each mode and uses those distances to recover nonlinear geometric structure.
> >
> >
> >
> > I agree that tensor decompositions and TCML formulate different objectives. However, tensor decompositions are now well-established and widely used across a broad range of tasks, including dimensional reduction. Their factor matrices or latent representations can be used as embeddings. Again, if the authors require a nonlinear setting, then nonlinear tensor decomposition needs to be considered. The current introduction claims "complementary setting", which does not establish the necessity of TCML.
> >
> >
> > > The experimental results show cases where this distinction is important, and using TCML outperforms tensor decomposition.
> >
> > In the current experimental setup, it is difficult to determine whether the gain comes from non-linearity or multiscale distances.
> >
> > The authors mention the elbow method in the rebuttal, but I could not find a sufficiently detailed rank-selection protocol in the revised manuscript. This is particularly important for Tucker and TT decompositions, where the rank is not a single scalar but a tuple of multilinear ranks or TT-ranks. The candidate rank sets, selection criterion, selected ranks, and whether the same protocol was applied consistently across datasets should be reported. In addition, the ordering of tensor modes can affect TT decomposition, but I could not find an elaborated discussion of how the ordering was chosen or whether alternatives were considered. In practice, these choices can substantially affect the performance of tensor factorization methods, so they should be reported.
> >
> > In my view, these issues are not minor presentation problems but concern the positioning and empirical support of the paper. I therefore do not think my main concerns have been sufficiently addressed within this revision.
> >
> >
> > The following are minor points and do not strongly affect my final recommendation, but I expected them to be fixed as well:
> >
> > - The color coding in Figure 1 is still not explained.
> > - The equation referencing style remains inconsistent, e.g., around page 5.

---

> ### Author Response · Authors · 2026-06-20
>
> Thank you for the follow-up discussion. We appreciate the feedback. We have uploaded a new manuscript which includes several improvements to address feedback. We will continue to put in effort to improve the manuscript further as you suggest. We would also like to highlight our algorithmic contribution, even apart from the embedding comparison, that the sparse-graph ADMM solver and compression strategy provides a provably convergent and reusable algorithm for nonconvex graph-regularized tensor smoothing/co-clustering.
>
> **Additional experiments with Nonlinear Kernel CP Decomposition** We have added a nonlinear tensor-decomposition baseline to the YaleFaces experiment. The kernel-subspace CP is based on the technique described in the paper [1]. For each mode $d$, we build an RBF kernel on mode-$d$ tensor slices, retain the leading centered kernel eigenvectors $Q_d$, project
> $$
> =X\times_1 Q_1^\top\cdots\times_D Q_D^\top,
> $$
> fit CP to $Y$, and lift the factors back as $A^{(d)}=Q_dB^{(d)}$. This gives one nonlinear CP decomposition. The results help to distinguish between gains due to nonlinearity of the embedding and from gains due to TCML’s multiscale construction. We note that there is no canonical missing-data extension for the simple kernel-subspace CP baseline we are adding. In particular, the kernel bases $Q_d$ require mode-wise slice similarities, and these similarities become unreliable under heavy missingness which may explain the degradation in performance at nonzero noise levels. In contrast, TCML has missingness built into the smoothing objective and performs imputation as multiple scales through the learned smooth tensors.
>
> **Parameterization of tensor-decomposition baselines and separating nonlinearity from multiscale distances**
> We also agree that the paper should not claim “necessity” in an absolute sense. We have revised the claim in the introduction:
> __TCML is useful when the target is a coupled mode-wise geometry.__
>
> This is distinct from searching for factor matrices associated with a primary objective of approximating data tensor values via a reconstruction loss. TCML instead constructs a coupled multiscale distance matrix for each mode. These distances can then be passed to existing nonlinear manifold-learning tools, such as Diffusion Maps, t-SNE, or UMAP. In this sense, TCML provides a tensor-aware interface to nonlinear manifold learning: it allows standard distance-based embedding methods to be used while retaining multiway coupling. In contrast, and as we show in the manuscript, applying Diffusion Maps directly to a matricized mode uses only a single-mode slice distance; this loses the cross-mode smoothing information, which is why the single-mode DM baseline fails to recover some coupled structures in our synthetic experiments.
>
> We also agree that the original experiments did not fully separate two effects: nonlinearity and multiscale distance aggregation. As you recommended, we have added another comparison to address this. We have add a nonlinear tensor-factorization baseline, kernel-subspace CP. This baseline gives nonlinear factor embeddings, but it does not perform TCML’s joint multiscale smoothing or multiscale distance aggregation. It therefore tests whether nonlinear factor coordinates alone explain the improvement.
>
> **Elaboration on CP rank selection** The exact choice of rank for the CP decomposition depends on the degree of missingness / noise. Generally, a lower rank degrades reconstruction loss, but yields an embedding that is more robust to missing values. We can identify a good rank selection by plotting this tradeoff and selecting ranks manually. Similarly, the TT and Tucker, we tune the multilinear and TT-rank tuples. We have included the CP decomposition, TT, and Tucker results on the Linkage2 dataset in the supplement (A.4 Figure 8) and will expand upon these results to other datasets if accepted. We note that 1) because TT and Tucker ranks depend on the ordering of tensor modes, we enforce the natural semantic ordering of the dataset when available 2) For the purposes of visualization, we present a simplified scalar sweep across the rank-selection procedure i.e., for Tucker we sweep ranks ranks (r,r,r) for each mode and for TT we sweep ranks (1, r, r, 1). We will expand upon this in the revised manuscript.
>
> **Additional minor revisions** We have also resolved the second minor issue noted: We have also reviewed equation references and have made them consistent throughout the revised manuscript.

---

> > ### Comment · Reviewer_y4M4 · 2026-06-23
> >
> > Thank you for the update. I appreciate the authors’ continued efforts to improve the manuscript. I have already added my final comments on the latest revision to the recommendation submitted to the AC.

---

### Decision · Action_Editor_nQBK · 2026-06-29

**Recommendation:** Reject

**Audience:**

Yes

**Audience Explanation:**

This submission has the potential to be valuable to the TMLR audience, while its contribution is not sufficiently demonstrated in the current manuscript. I believe it could become a valuable contribution after a major revision that properly addresses the concerns raised by the reviewers.

**Claims And Evidence:**

No

**Claims Explanation:**

As the reviewers pointed out, this issue is unfortunately not addressed sufficiently. A careful comparison with existing tensor-based dimensionality reduction methods is essential to support the main claims of the paper. Addressing this concern would require a substantial revision, and therefore I recommend rejection at this stage.

I would also expect a more thorough explanation of the graphs used as regularizers. They appear to play a central role in the proposed regularization framework, yet it remains unclear how they are constructed, why they are introduced, and how they are integrated with the tensor representation. In particular, the motivation for introducing the graphs is insufficient, for example, the Introduction provides virtually no discussion of their role or necessity.

**Resubmission Of Major Revision:**

The authors may consider submitting a major revision at a later time.